# Assessing population exposure for landslide risk analysis using dasymetric cartography

Ricardo A. C. Garcia[1], Sérgio C. Oliveira[1], José L. Zêzere[1]

[1]Centre for Geographical Studies, Institute of Geography and Spatial Planning, Universidade de Lisboa, Lisboa, 1600-276, Portugal

*Correspondence to*: R.A.C. Garcia (rgarcia@campus.ul.pt)

**Abstract.** Assessing how many people are exposed and their location is a crucial step in landslide risk management and emergency planning. Frequently the available population statistical data have insufficient detail for an accurate assessment of the potentially exposed people to a hazardous phenomenon, mainly if it occurs at the local scale, such as landslides. The present study aims to apply the dasymetric cartography technic to improve population spatial resolution and to assess the potentially exposed population. An additional objective is to compare the results with those obtained with a more common approach that uses, as spatial units, the basic census units which is the best spatial data disaggregation and detailed information available for regional studies in Portugal. Considering the Portuguese census data and a layer of residential building footprint, whose area was used as ancillary information, the number of exposed inhabitants differs significantly according to the approach used. When the census unit approach is used and considering the three highest landslide susceptible classes, the number of exposed inhabitants is in general overestimated. Despite the associated uncertainties in a general cost-benefit analysis the presented methodology seems to be a reliable approach as first approximation to a more detailed estimation of exposed people. The approach based on dasymetric cartography allows increasing the spatial resolution of population over large areas and enables the use of detailed landslide susceptibility maps, which is a major added value for improving the exposed population assessment.

Keywords: people exposure, people spatial distribution, dasymetric, cartography, landslides

## 1 Introduction

### 1.1 General concepts and framework

In natural sciences, risk is a function of the probability of occurrence of a hazard scenario and the related consequences that are expected on the exposed elements at risk (e.g. Varnes and IAEG, 1984; Fuchs et al. 2013). So a complete landslide risk analysis is a function of the frequency and process magnitude (e.g. Guzzetti et al., 1999), and of the level of damages and

associated costs (e.g. Varnes and IAEG, 1984). However, in some cases it not possible to quantify the time recurrence and/or the landslide magnitude and the susceptibility considered as the likelihood of landslide occurrence in a specific area according to terrain conditions (Brabb, 1984), can be used as a first and simple approach of phenomena occurrence (e.g. Guillard-Gonçalves et al., 2015).

Regarding the vulnerability assessment, different vulnerability dimensions (e.g. social, personal, structural, economic, political and environmental) are frequently taken into account (Fuchs, 2009; Kienberger et al., 2009) and linked to each other (Fuchs 2009; Papathoma-Köhle et al. 2011; Kappes et al. 2012). Therefore, integrated approaches to assess vulnerability have gained popularity in recent years (Fuchs et al., 2011, Karagiorgos et al., 2016); nevertheless they require a quantitative evaluation of each vulnerability component, such as the assessment of elements at risk, their physical exposure and social
characteristics (Karagiorgos et al., 2016). However, the variety of potentially exposed elements, i.e. elements present in hazardous zones that are thereby subject to potential losses (e.g. UNISDR, 2009), and their different characteristics (e.g. buildings, roads, people) leads to a complex and multi-level analysis; consequently studies that include more than one type of exposed elements are scarce (e.g. Michael-Leiba et al., 2003; Keiler, 2004; Promper and Glade, 2016).

Frequently only two vulnerability dimensions are considered: physical vulnerability and social vulnerability. Regarding the
physical vulnerability dimension attempts have been made to establish empirical relationships between process intensity, ,type and number of exposed elements, in order to estimate expected degree of loss (e.g. Galli and Guzzetti, 2007; Papathoma-Köhle et al., 2007, 2012; Petrucci and Gullà, 2010; Kappes et al., 2012; Silva and Pereira, 2014; Uzielli et al., 2014; Winter et al., 2014; Fuchs et al., 2015; Promper et al., 2015; Guillard-Gonçalves et al., 2016).

Concerning the social sciences, attention has been drawn to the way communities and society in general cope with and adapt
to disaster events (e.g. Cutter et al., 2003; Kienberger et al., 2009; Mendes, 2009; Nathan et al., 2010).

Other studies have tried to evaluate the relationships between process occurrence and injuries to people and/or their evacuation, by calculating the probability of fatalities and their acceptability/tolerance, and combining approaches to build f-N curves (e.g. HSE, 1992; Cruden and Fell, 1997; Evans, 1997; Guzzetti, 2000). Further studies have evaluated the probability of people being affected outside or inside an element (e.g. a house) that is hit by the hazardous phenomenon (e.g.
Ragozin and Tikhvinsky, 2000; Bell and Glade, 2004; Kaynia et al., 2008). The abovementioned studies are generally based on historical data of hazard phenomena that affected population (e.g. Dai et al., 2002; Guzzetti et al., 2005). However, these historical databases are often insufficient and incomplete, which means that "probabilities" have frequently been based on knowledge and judgement (Michael-Leiba et al., 2003).

Truly interdisciplinary research is needed for the analysis of such dynamical and complex topic (Fuchs et al., 2011). In
addition, different datasets of elements must be taken into account (e.g. building structure and materials, number of inhabitants, infrastructures uses, traffic volume, among others) to estimate direct and indirect costs within the quantitative risk analysis (e.g. Zêzere et al., 2007, 2008; Remondo et al., 2008; Corominas et al., 2014; Schwendtner et al., 2013). Conversely, the lack of interdisciplinary and multi-level approaches (e.g. regional/international, personal/political) can reduce the efficiency of adopted policies designed to avoid disasters (e.g. Xanthopoulos, 2007; Aubrecht et al., 2013).

The accurate assessment of "how many people" are present at a certain time and place is crucial for emergency planning namely to manage people evacuation. To know, as precisely as possible, the location of potentially exposed persons is mandatory to guarantee the efficiency of emergency plans and to reduce associated costs related with the rescue of people and social recovery (e.g. Bhaduri et al., 2002; Sutton et al., 2003; Chen et al., 2004; Su et al., 2010; Freire and Aubrecht, 2012; Freire et al. 2012; Aubrecht et al., 2013). In fact, according to Bhaduri et al. (2002) locating population at risk must be the first step in saving lives.

The accurate knowledge of the number and location of exposed persons is mandatory for a complete risk analysis that further calls for the harmonization between the resolution of the hazard and detailed population data distribution. A high resolution of population distribution is mainly needed when the hazard has no extensive consequences, as in the case of landslides, where the processes are more selective and local damage related (Deichmann et al., 2011). Additionally, in larger study areas where diverse types of occupation can take place (urban, rural) significant differences on population density are expected to exist. When the combination of these two situations occurs (local hazard and sparse population) it becomes even more important to know precisely the location of the exposed people.

## 1.2 Assessment of population exposure - state of the art

Vulnerability of potentially affected people are usually assessed based on inhomogeneous spatial units, such as the municipality or the parish (e.g. Santos et al., 2014; Guillard-Gonçalves et al., 2015), and on census data which mainly indicates night-time distribution of the population (e.g. Freire and Aubrecht, 2012; Aubrecht et al., 2013; Fraser et al., 2014; Tavares and Santos, 2014). For this reason some authors tried to evaluate the population fluctuation (daily, seasonal, historical) in order to assess the distribution of exposed people (e.g. Keiler, 2004; Keiler et al., 2005; Freire and Aubrecht, 2012; Schwendtner et al., 2013). Aubrecht et al. (2010) provided a detailed approach, in a 13 km$^2$ area, by adding to a high resolution land cover map information about building height (as proxy of building capacity) and building use (residential, public, commercial, others). Freire et al. (2012) used a 3D building model to estimate how many people would need to be evacuated from a 2.5 km$^2$ estuarine area in Lisbon in case of a tsunami. Fuchs et al. (2015) assessed exposure to several hazard phenomena in Austria based on unusually detailed property data information, as for example: height of buildings, net area, configuration, main usage, and number of people per building. Although, quite detailed methods for disaggregation of people and for counting the number of people exposed to a hazard have already been tested, the need for high detailed information does not allow its widespread use in areas of hundreds of square kilometres.

Therefore, the scale of analysis, as a proxy of data detail, is a major support to vulnerability assessment, namely when assessing people's exposure. Although overall vulnerability models are consistent at different scales, the aims and variables that drive an analysis at municipal scale are different from those used at the town scale, whereby a "re-scaling" of

approaches is essential (Mendes et al., 2010; Tavares and Santos, 2014; Tavares et al., 2015). Additionally, there are countries like Portugal where, due to privacy policies, the best source of population data (e.g. census) is only available with aggregated information which distorts reality. Even in the smallest census units as the Basic Census Unit (BCU) homogeneity is not always achieved. In these cases, the assumption of homogeneity leads to an error that increases with the

diversity of uses (e.g. residential, commercial and agricultural). At best the BCU corresponds to city blocks in the urban areas in Portugal but it may have a huge variation in size in sparsely populated rural areas, which constrains the accurate assessment of the location of people exposed. Even if the data were collected individually an aggregation would be done which implies the assumption of a uniform distribution of people inside the aggregation unit, i.e. population could be distorted (e.g. Fisher and Langford, 1996; Su et al., 2010). In addition, people can be concentrated in specific places within a

BCU. Therefore, a better resolution of population data is needed.

In order to obtain a finer spatial distribution of the population, different methods and data can be used, mainly when considering different scale approaches (Aubrecht et al. 2013). The adopted methodology and obtained results are dependent on the type and quality of the input data used as ancillary information to disaggregate data (Su et al., 2010). In fact, on global scales (from world to regional scale), the tool to disaggregate general information is frequently a land use map or an

accessibility map that allows a spatial discrimination of the population between urban and rural areas to be made (e.g. Eicher and Brewer, 2001; Mennis and Hultgren, 2006; Reibel and Agrawal, 2007; Langford, 2007; Langford et al., 2008; Gallego, 2010; Steinnocher et al., 2011). The drawback of these approaches is the limited spatial resolution of the land use map that leads to overestimation or underestimation of the population in sparsely populated areas (Steinnocher et al., 2011; Aubrecht et al. 2013). At local scales (municipality to parish), due to the detailed input data, it is possible to consider urban systems

with finer grid cells and to take into account as weighing factors, parameters such as built-up areas, roads typology or population fluctuation (e.g. Keiler, 2004; Reibel and Bugalino, 2005; Freire and Aubrecht, 2012; Fuchs et al., 2013).

## 1.3 Objectives

In this framework, the major aim of this work is to develop a methodology to assess population exposed to deep rotational slides and to increase the population resolution over large areas. Population exposure is based on a detailed landslide susceptibility map (pixel 5m) (Garcia, 2012), the resident population from national official statistics (Census 2011), and a building footprint layer, as ancillary information. A major goal of the current work is to assess the distribution of people over the various buildings. This can be considered as an intermediate and quick approach between coarser assessments (e.g.

parish level) and detailed and time-consuming local approaches. Additionally, the differences between exposed people are assessed by comparing a more traditional approach (considering population per census units) and using a distribution of population per building. This finer distribution is based on dasymetric mapping, which is a cartographic technique that uses

ancillary information to increase the resolution of the coarser input data. The present study is applied on the Alenquer river basin which is located to the North of Lisbon (Portugal).

## 2 Study area

The study area is the Alenquer river basin (120 km$^2$), which is located north of Lisbon (Fig. 1). The choice of this study area
was based on three reasons: i) landslides incidence; ii) type of urban occupation; and iii) social vulnerability.

i) The study area is located to the north of the Lisbon region that is a landslide prone area (Zêzere et al., 2008) and according to the DISASTER database (Zêzere et al., 2014) is one of the areas in Portugal that has sustained severe landslide damage. The present work focuses only on deep rotational slides (depth of rupture zone > 3 m). These landslides are generally slow but encompass horizontal displacements capable to significantly damage structures (e.g. houses) and consequently entail
evacuation of people (Garcia, 2012);

ii) The study area presents two types of "urban landuse": small villages with a dense urban grid and disperse settlements. The Census units boundaries were influenced by settlements density, therefore the existence of two different types of territorial occupation in the study area allows the comparison of the proposed methodology applied to two different "urban" contexts;

iii) The study area is, theoretically, one of the least prepared to deal with landslide consequences within the region north of Lisbon. According to Mendes et al. (2010) that evaluated the social vulnerability at the municipal scale in Portugal, the Alenquer municipality has a medium criticality ("defined as the ensemble of individuals' characteristics and behaviours that may contribute to the system's rupture") and low capability ("defined as the set of territorial infrastructures that enables the community to react in case of disaster").

The elevation in the study area ranges from 20 to 375 meters and the major landforms are hills and fluvial valleys, which are strongly controlled by differences in resistance of the bedrock, such as sandy-marl (particularly prone to rotational landslides), sandstone and limestone. Field work and aerial photo interpretation allowed the identification and mapping of 136 rotational slides (0.98 landslides/km$^2$) that generated a total unstable area of 663,508 m$^2$ (0.56 % of the study area) (Garcia, 2012).

Concerning human occupation, the study area has 15,253 inhabitants (Census 2011) most of whom live in the village of Alenquer located in the SE sector of study area (Figure 1). The remaining population lives in small villages scattered in an area where agriculture is the dominant activity. Cadastral cartography and field work identified over 6,889 residential buildings that were included in this work. Considering the Basic Census Unit (BCU as the best Census spatial resolution available for population data, the area is covered by 676 BCU with a wide range of surfaces (minimum: 280 m$^2$; mean:
176,100 m$^2$; maximum: 4.4 km$^2$). The mean BCU population is 26 inhabitants (disregarding the 10 % BCU that have no inhabitants) and the maximum population per BCU is 357 inhabitants.

**3 Data and methodology**

The most detailed public information about population available in Portugal comes from the national census Basic Census Unit (BCU), in which the smallest territorial units correspond to city blocks. However, these units are inhomogeneous in space, and consequently in number of buildings and inhabitants, namely in rural areas or transition areas between urban and rural, that form the study area.

The general methodology to assess exposed population in two different terrain units follows three main steps (Fig. 2):

(i) the assessment and classification of landslide susceptibility for both spatial units (pixels and BCU). When using BCU units, it is mandatory to employ a generalization technique;

(ii) the evaluation of population distribution considering the different spatial entities (BCU and target zones within BCU);

(iii) the integration of susceptibility and population distribution in order to calculate the potentially exposed inhabitants in each susceptibility class based on different spatial entities (referred in (ii)).

**3.1 Landslide susceptibility**

Landslide susceptibility was assessed at the pixel level using the Information Value method (Yin and Yan, 1988), which is a Bayesian bivariate statistical model that has been shown to be suitable for landslide susceptibility assessment (e.g. Piedade et al., 2011; Guillard and Zêzere, 2012, Pereira et al., 2012; Oliveira et al., 2015 and references therein); it has further been recommended as a method for data-driven landslide susceptibility assessment worldwide (Corominas et al., 2014). The susceptibility assessment procedures were based on the work of Garcia (2012), namely, landslide inventory based on field work and interpretation of aerial photo with 0.5 m resolution. The landslide database includes only deep rotational slides (rupture surface deeper than 3m) that were divided in two independent groups based on temporal criteria, one used for modelling landslide susceptibility, and the other used for the independent validation of the landslide susceptibility model. The landslide modelling group includes all the rotational slides that occurred up to the regional landslide event of March 2010 (Zêzere and Trigo, 2011) (104 cases) and the landslide validation group includes all the rotational slides that occurred during that landslide event (32 cases).

Six landslides predisposing factors were use as independent variables: slope, lithology, land usage, inverse of wetness index, morphostructural units and soil type. Lithology, soil type and land use were based on national official cartography at 1:25 000 scale. The slope and the inverse of wetness index were derived from a digital elevation model (DEM) built based on a 5 m contours topographical map. The morphostructural units map was obtained by combining the aspect map derived from the DEM with information on dipping direction of lithological layers obtained from geological maps and field work.

The susceptibility model was further validated using success rate curve, prediction rate curve (Chung and Fabbri, 2003, 2008) and calculating the area under the curve (AUC) (Sweets, 1988).

The final susceptibility model was made with a 5 m resolution pixel and was classified using five quantile classes, i.e. each landslide susceptibility class includes 20% of the study area. The option for the classification based on a quantile method

aims to get susceptibility classes with similar size without assigning importance *a priori* to any of those classes based on their sizes. However, to use census unit maps a susceptibility value should be addressed to each BCU. So, in a subsequent step the classified pixel-based landslide susceptibility map was overlaid to the BCU map (vector structure), and a landslide susceptibility classification attributed to each BCU. The generalization was made using two different techniques: i) the BCU susceptibility class was defined according to the majority landslide susceptibility class presented in the BCU; ii) the overall susceptibility of the BCU is the weighted average of identified susceptibility classes.

## 3.2 Population distribution and exposure

The potentially exposed population to landslide risk was assessed using the Census (2011) data and two approaches: (1) to take into account the population within each BCU (residential population); and (2) to distribute the population by the residential buildings within each BCU using dasymetric cartography.

Dasymetric cartography is a classic approach (Wright, 1936) that has recently been used as an analytical tool based on Geographical Information Systems (e.g. Eicher and Brewer, 2001; Mennis and Hultgren, 2006). The dasymetric cartography use ancillary information to increase the resolution of coarser input data. A set of target zones should be defined and then, based on areal interpolation or other weighting algorithms, the input data should be disaggregated to estimate, for example, the population in a set of smaller units based on the known population for the global unit (e.g. Flowerdew and Green, 1992; Langford and Unwin, 1994; Mennis, 2003; Holt et al., 2014; Wu et al., 2008; Su et al., 2010; Tapp, 2010). In this work the dasymetric approach was performed following a binary analysis over residential building/not residential building areas (Fig. 3). The building layer (1:10 000 vector map from Alenquer Municipality) has attribute fields that allows differentiating the type of services and commercial buildings (e.g. police stations, fire stations, schools, court, medical facilities, among others). Additionally, during detailed field work the non-residential buildings were identified, e.g. storage buildings, factory buildings, and that information was added to the original database. All the other buildings were regarded as intended for residential use. However, some buildings could have more than one function. In the present work all the buildings that were exclusively residential or mainly residential were considered as ancillary information. The remaining buildings were not considered as target zones and they were not assigned any population.

Thus, the first step is the definition of target zones. In this work a layer with the residential buildings was used. Disaggregation methods, based on cadastral information are the best approach to a realistic population distribution/location (Maantay and Maroko, 2009). By overlaying BCU and buildings layer (both in a vector structure) it is possible to identify the potentially inhabited areas (target zones) in each statistical unit.

Two different population densities were calculated, considering the target zones and the BCU. To compare the obtained results within BCU and target zones, density maps were classified accordingly to standard deviation method.

The second step was the weighting of each target zone, i.e. the importance of each building ($W_{tzi}$) inside a specific BCU. In the present work, the area of the building was the sole parameter considered for weighting the target zone importance

because the available Census (2011) data are aggregated at BCU level and the layer of the ancillary cadastral information (buildings) only has the footprint, disaggregated for each individual house.

The third step is the dasymetric distribution of the population in each polygon of target zones ($P_{tzi}$) as show in Eq. 1 (adapted from Su et al., 2010):

$$P_{tzi} = \frac{P_t \times W_{tzi}}{\sum_{i=1}^{n} W_{tzi}} \tag{1}$$

where $W_{tzi}$ is the weight of each target zone in the BCU and $P_t$ is total population in the BCU. This procedure was applied independently to each BCU to distribute the population among the buildings in each unit. After disaggregation the total number of inhabitants per BCU is maintained.

The last step was the assessment of the number of people exposed in each landslide susceptibility class. For this purpose the integration between the susceptibility map and population distribution is needed. In the case of BCU as terrain units, the assessment is direct because each BCU is classified within a single susceptibility class (Sect. 3.1) and has a population assigned by the Census. On the contrary, if target zones (buildings) are used as terrain units, parts of a single building can fall into different susceptibility classes. In these cases it was necessary to convert the target zones from vector to a grid structure, consistent with the 5 m resolution susceptibility map. The population in each building is then equally distributed among the pixels that cover that building. For example, a 100 m$^2$ building (converted in 4 pixels of 5 by 5 meters) that has 8 inhabitants will have in the final population distribution 2 inhabitants in each pixel. The conversion of the target zones from vector to a grid structure was only made to assess the number of people exposed in each susceptibility class and therefore to easily compare the results obtained among approaches. So, for practical purposes the population is assigned to each building and not to a pixel.

Finally, the assessment of the number of inhabitants in each susceptibility class was performed for three different scenarios: i) the susceptibility map was generalized to BCU according to the majority class and all the BCU population was assigned to that specific susceptibility class (approach 1a); ii) the susceptibility map was generalized to BCU according to weighted mean of susceptibility classes in that unit and all the BCU population was assigned to that specific susceptibility class (approach 1b); and iii) the susceptibility map is in pixel units (without generalization) and the population (dasymetricly distributed) was assigned to the susceptibility class of the pixel (approach 2).

## 4 Results

The landslide susceptibility map (pixel terrain unit) validation yields acceptable results with a 0.76 AUC for the success rate curve and 0.78 AUC for the independent validation with the prediction rate curve.

Figure 4 shows the obtained landslide susceptibility maps using different terrain units and generalization techniques: (a) pixel based map; (b) BCU vector structure map according to the dominant susceptibility class inside each BCU and (c) BCU vector structure map according to the weighted average susceptibility class in each BCU.

The visual differences among maps are evident, mainly between the BCU susceptible map classified with the weighted average susceptibility (Fig. 4c) and the other maps. In fact, the use of the weighted average generates a significant decrease of the importance of the extreme classes (Very high and Very low). The differences between the two generalization methods are significant (Fig. 4b and 4c) showing a five-fold increase in the importance of the moderate class in approach 1b (Tab. 1). However, the visual agreement between maps 4a and 4b is evident. Although the homogenization of the susceptibility classes per BCU leads to an increase of about 8% of the area classified with very high susceptibility (Tab. 1), when the three highest susceptible classes are combined their overall representation remains equal to the pixel map model (60% of the total study area). Conversely, the same 3 susceptibility classes extend to 83% of the total study area in the weighted average susceptibility map, namely due to the over representation of the moderate class. When evaluating population densities the use of different spatial units (BCU and BCU built-up area) shows, as expected, considerable differences (Fig. 5). In fact, if a common approach is adopted and BCU are classified according to their overall area (population density per BCU area), density values (mean: 0.002 inhabitants/m$^2$; SD: 0.003) are around one order of magnitude lower when compared with results obtained considering only the built-up area (population density per BCU residential building area) (mean: 0.011 inhabitants/m$^2$; SD: 0.009). Additionally, Fig. 5 shows that in high building density terrain units (blue outline squares example), the registered population density hierarchy remains similar in maps (a) and (b), whereas in rural terrain units (red outline squares example) differences can be considerable (more than two standard deviations in the example shown). These differences are relevant in areas similar to the ones in the case study where the majority of BCU (73 %) have a residential built-up occupation below 20 %, which means that the use of total BCU area generates the underestimation of the population density.

Table 2 shows the results obtained regarding the number of potentially exposed population per susceptibility class considering the three different approaches (1a, 1b and 2). It is clear that the number of exposed inhabitants changes considerably depending on the method used to estimate the population. The approaches 1a and 1b, in general, systematically generate a higher number of inhabitants in the three most susceptibility classes than approach 2. The only exception is in approach 1b wherein the population assigned to the very high susceptibility class has only 31 inhabitants, which is explained by the diminished importance of this susceptibility class when generalization is based on the average susceptibility. In fact, when the generalization of the susceptibility map is carried out, the number of exposed people is 29 % (approach 1a) and 35 % (approach 1b) of the total population. In contrast, when the most detailed susceptibility map is used, allowing the use of the dasymetric distribution of population, the number of exposed inhabitants is much smaller (1926 people, 13% of total inhabitants). The number of people exposed in the three most susceptible classes, within approach 1a, exceeds 132% features of approach 2, which means that 2539 inhabitants are overestimated when using approach 1a.. In addition, for practical use

in emergency management, approach 2 allows the cartographic expression of people per building (Fig. 6), which is not the case of approaches 1a and 1b.

## 5 Discussion

In this work, three different approaches were used to evaluate the potentially exposed inhabitants in a test site located in the Alenquer municipality.

The use of Census, as source of population data, requires two major assumptions: i) the resident population does not change in time; ii) people are located at home. These are strong assumptions in the sense that residents are presumed to be at home at all times, and that it does not take into account the fact that people living outside the study area might actually be in the study area. In fact, this is far from reality because people move around during the day. However, in what concerns the study area there are no data about daily or seasonal fluctuation of population neither at the building scale nor at the considered statistical unit. So, the above scenarios can be considered as the worst case scenarios for the resident people but the fluctuations during day/night to work, school or other outdoor activities should not be neglected. Additionally, the use of the worst case scenario is supported by the work of Pereira et al. (2015) which found that, in the period 1865-2010, the majority of landslide fatalities occurred while people were indoors (60%) whereas 40% occurred when people were outdoors or in a vehicle.

Once the population data is available in statistical units the use of these data implies the generalization of the landslide susceptibility map from the raster structure to the statistical unit. Conversely, the approach that considers target zones (buildings) within each BCU to distribute population enables the use of the original landslide susceptibility map with a 5m pixel.

Independently of the approach some uncertainties are present and affect the obtained results. In fact, the classification of the "original" landslide susceptibility map (pixel-based) is needed to generate landside susceptibility maps based on statistical units. The number of classes and the chosen method to generalize the susceptibility may produce differences in the obtained results once the range of classes may be higher and the importance of each class become significantly different, which influences the number of inhabitants in each class. However, the focus of this work is not to evaluate how classification methods or different number of susceptibility classes influence the assessment of exposed inhabitants. Hence the option for the classification based on a quantile method that aims to get susceptibility classes with similar size without assigning importance *a priori* to any of those classes based on their sizes. Moreover, the number of persons in each susceptibility class is only used to compare the adopted approaches. Even though the number of people per susceptibility class can change, it does not affect the distribution of people per buildings that is independent of the number of susceptibility classes.

Indeed, part of the differences observed on the exposed population from approach 2 to approaches 1a and 1b is due to the generalization process of landslide susceptibility per BCU, which can generate an overestimation or underestimation of the total area of each susceptibility class when compared to the pixel-based susceptibility map. As a consequence, the classification of the same building can be very diverse in the produced landslide susceptibility maps (pixel-based and BCU-based) (Fig. 7). In some few cases a building located in the very high susceptibility class in the pixel-based map is classified as very low susceptibility in the BCU-based map, due to large spatial expression of that class within the BCU terrain unit (Fig. 7b), thus producing an underestimated exposure to landslide hazard. However, in the majority of cases, buildings are located in a very low susceptibility class in the pixel-based map but due to the generalization of susceptibility they become included in the very high susceptibility class in the BCU-based map (Fig. 7a), thus producing an overestimated exposure to landslide hazard. The use of the majority class, as classification method, in the BCU susceptibility map is a source of error that tends to overestimate exposure of buildings and indirectly exposure of inhabitants.

On the other hand, the use of a weighted average classification tends to overemphasize the importance of the mean susceptibility class and underestimate the extreme susceptibility classes. When using statistical units susceptibility, an analysis should be done to the previous classification of the susceptibility map (in pixel structure) to evaluate the considerable changing area of the landslide susceptibility classes which will be reflected in the obtained results of people exposure. However, independently of the previous tests this kind of approach has always a large degree of generalization once it assumes that all spatial units are homogeneous in terms of landslide susceptibility.

Approach 2 is based on detailed pixel susceptibility map and does not require the generalization of landslide susceptibility which is a major advantage of this method. Approach 2 is a user-friendly methodology that allows improving the accuracy of the population spatial distribution and consequently improves the evaluation of the number of inhabitants exposed to a hazardous phenomenon. However, this approach is not free from uncertainties. The definition of target zones is one source of uncertainty. Therefore a binary classification that takes into account the residential use of the building was done. Despite the fact that the generality of the buildings have their use officially classified in the building layer database and field work validation had been carried out, not all the buildings were individually validated, which is a source of uncertainty. However, we consider that the errors associated to this uncertainty can be neglected due to three reasons: i) the majority of buildings have an exclusively residential use (93%) and the buildings that other than residential use have more than one type of use, are small in number (5% of total buildings); ii) the vast majority of the buildings (96%) in the study area have up to two floors; and iii) once only the area of the building is considered and "double" functions of buildings are confirmed usually in different floors of the building (e.g. ground floor - commercial, 1st floor - residential) the effective area considered as target zone is correct even if the ground floor is not for residential usage.

The weighting of each target zone is another source of uncertainty. In fact, equal building areas can have different number of floors, different number of bedrooms, and consequently a potentially different number of inhabitants, which is probably the major cause of uncertainty of this study. However, as already mentioned, 96% of the buildings are in the same class considering the number of floors (1-2), and buildings with higher number of floors are located in urban areas where the size

of the BCU are homogeneous and small in size. Therefore, we are confident that the achieved overall population distribution values are representative of the reality in the study area. Despite some uncertainties related to the distribution of people in the buildings, particularly in the more susceptible zones, this approach is definitely closer to reality than the one that views the total population as coming from BCU (the finest public population information available), once the susceptibility in BCU

units is far from being homogeneous.

In practical terms, the use of the approach 2 allows estimating the exposed people in each building and is cartography, which is important for risk analysis and emergency management. Although this can be considered a coarse distribution, because only the area was used to weight the importance of each building, this is a more detailed approach than the use of the total population per statistical unit.

Despite the described uncertainties, in real emergency situations where 3 or 4 identified buildings were affected by a landslide, the quick knowledge of the approximate number of potential victims is essential for the correct allocation of rescue resources.

In this work the people had to be considered always 'at home', which is a drawback for the analysis. In addition, the building resistance was not accounted for and the assessment of the exposed inhabitants is insufficient to demonstrate the real

exposure of people to landslide hazard. Topics as degree of people vulnerability due to their own characteristics (e.g. mobility, age, education and number of years living in the building) and due to existing infrastructures and facilities (e.g. sewerage, water or electricity supply, medical care, etc.) should be considered in a broader and more complete study.

Nowadays, it is assumed that the analysis of the vulnerability of the elements at risk may be the key to risk reduction (Papathoma-Köhle et al., 2016) and that detailed information on the characteristics and types of the current building

functionality, dimension and number of residents would enhance the significance of the results with respect to exposed citizens (Fuchs et al., 2015). Lastly, considering that people's physical vulnerability can be related to the fact that they are inside a building, a more detailed distribution of people inside buildings complemented with information on buildings resistance, such as construction materials, age or maintenance, would be essential to landslide risk management and to support the implementation of mitigation strategies. Summing up, for an integrated vulnerability study it is necessary that

physical damages (building, persons) but also functionality (e.g. infrastructures, services) and the social community as a whole be included. The already mentioned topics associated to the people fluctuation data, the improvement of the classification of building typology and the use of additional variables to weight target zones are forward working hypothesis to improve the obtained results.

**6 Conclusion**

From the point of view of a general cost-benefit analysis, the Census data (available and free of charge in Portugal) and the digital maps with building footprint (available or easily acquirable by digital image interpretation) taken as ancillary

information to dasymetric mapping approach prove to be a good option to increase resolution of the population distribution at the regional/municipal scale and it can be considered as a first approach to identify sites where future detailed surveys should be developed. Additionally it allows fast, partial (per BCU) or global, upgrades every time new information (e.g. population, building environment or landslide susceptibility) is provided.

Thus, the methodology developed using dasymetric cartography for the population distribution reveals three main advantages: (i) the increase in population resolution which allows a more detailed evaluation of the number of inhabitants potentially affected by a hazardous event; (ii) the increase in population resolution allows the use of detailed susceptibility maps avoiding generalization procedures that cause undesired homogenization of the data; and (iii) the location of people is confined to an area (buildings) with physical limits (not administrative nor statistical) that can be easily recognized by those

responsible for civil protection, planning and emergency management; such is not the case when the analysis is performed using a grid cell-based map. However, some uncertainties related to the dasymetric population distribution are present, generally associated to three main assumptions that have to be adopted: i) the binary classification of the use of the building (residential/not residential); (ii) people are always inside the buildings; and (iii) the building area was considered as the only proxy of the number of inhabitants per building.

The proposed methodology can be applied in multi-hazard studies and it is useful in both situations considering probability-intensity relations: (i) low probability phenomena and high magnitude that can result in high level of damages, and (ii) high probability events and lower magnitude that are expected to result in few affected elements. In both cases, the estimation of the number of inhabitants per building will be useful to increase the efficiency of actions taken by the Civil Protection. In fact, the prioritisation of buildings bearing in mind the potentially affected inhabitants will enhance the accuracy of rescue

operations. In case of events that cause generalized damages over a large territorial extension the focus on a specific building will not be so important because the whole region is affected. The exception can occur in low density urbanization areas and in the buildings where a high concentration of people is expected. In case of magnitude/high frequency events, local damages gain importance and therefore the proposed approach can be more useful. However, this understanding is completely dependent on the type of process, elements at risk, Civil Protection procedures, among many other factors that

influence emergency management operations.

Lastly and in spite of our good results, we would like to point out that assessing exposure of inhabitants is just a first step towards a desirable, integrated vulnerability analysis and a complete risk analysis.

**Author contribution**

R.A.C. Garcia and S.C. Oliveira performed field work for landslide inventory and cartography data base validation. Cartographical and statistical input data, susceptibility modelling and dasymetric cartography adaptation were carried out by

R.A.C. Garcia that prepared the manuscript with contributions from all co-authors (that supervised all the work development)

**Acknowledgements**

The authors would like to thank Dr. Maria Papathoma-Köhle and Dr. Alexandre Tavares whose deep review, constructive comments and suggestion contribute to improve the quality of the final manuscript.

This work is part of the project FORLAND - Hydro-geomorphologic Risk in Portugal: Driving Forces and Application for Land Use Planning (PTDC/ATPGEO/1660/2014) financed by the Portuguese Foundation for Science and Technology (FCT). S.C. Oliveira obtained a Post-doctoral grant [SFRH/BPD/85827/2012] from the Portuguese Foundation for Science and Technology (FCT).

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

**Table 1.** Landslide susceptibility classes (%) in Alenquer study area considering Approach 1a (BCU susceptibility generalization considering the majority susceptibility class), Approach 1b (BCU susceptibility generalization considering the weighted average susceptibility) and Approach 2 (pixel-based susceptibility map).

| Susceptibility class | % of study area | | |
| --- | --- | --- | --- |
| | Generalized BCU susceptibility maps | | Pixel susceptibility map |
| | Approach 1a | Approach 1b | Approach 2 |
| | Majority value | Average value | |
| Very high | 28.62 | 0.22 | 19.96 |
| High | 19.63 | 23.92 | 19.98 |
| Moderate | 12.01 | 59.09 | 19.98 |
| Low | 21.51 | 13.36 | 20.08 |
| Very low | 18.23 | 3.41 | 20.00 |

**Table 2.** Potentially exposed population per susceptibility class in Alenquer study area considering Approach 1a (BCU susceptibility generalization considering the majority susceptibility class), Approach 1b (BCU susceptibility generalization considering the weighted average susceptibility) and Approach 2 (pixel-based susceptibility map).

| Susceptibility class | Potential exposed population | | |
|---|---|---|---|
| | BCU population distribution | | BCU population per building |
| | Approach 1a | Approach 1b | Approach 2 |
| Very high | 1,840 | 31 | 430 |
| High | 1,201 | 1,230 | 675 |
| Moderate | 1,424 | 4,197 | 821 |
| Low | 1,639 | 4,692 | 1,454 |
| Very loz | 9,149 | 5,103 | 11,873 |

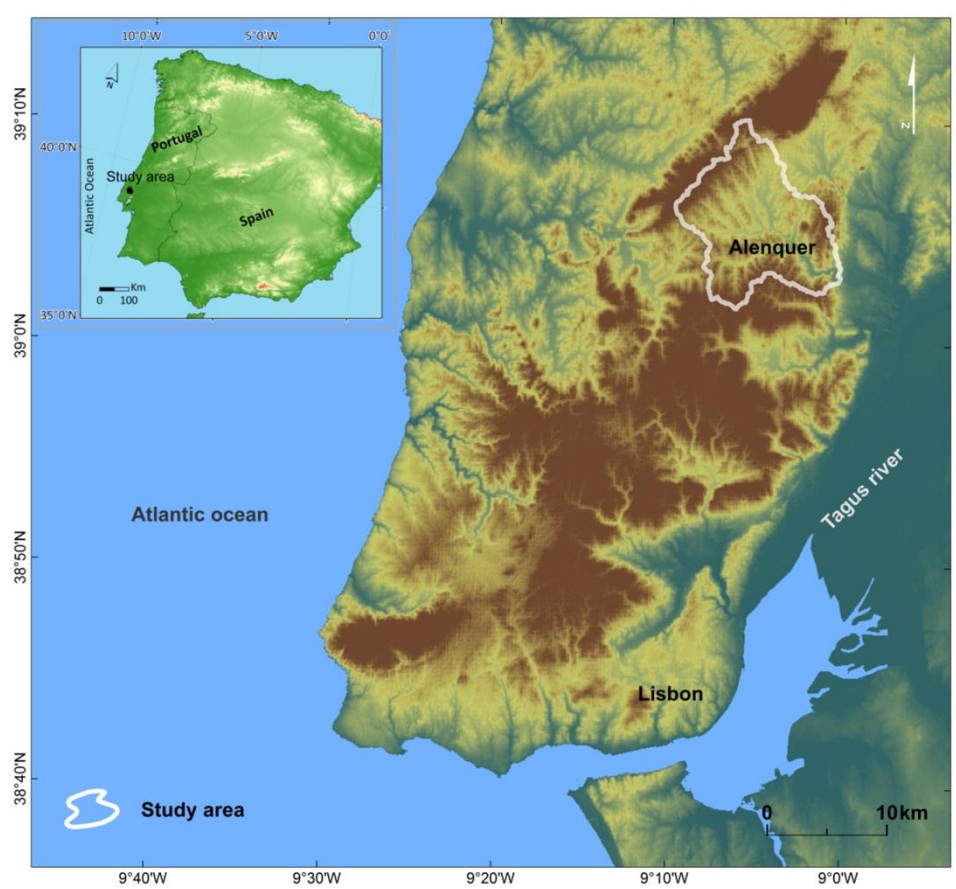

**Figure 1: Location of Alenquer study area.**

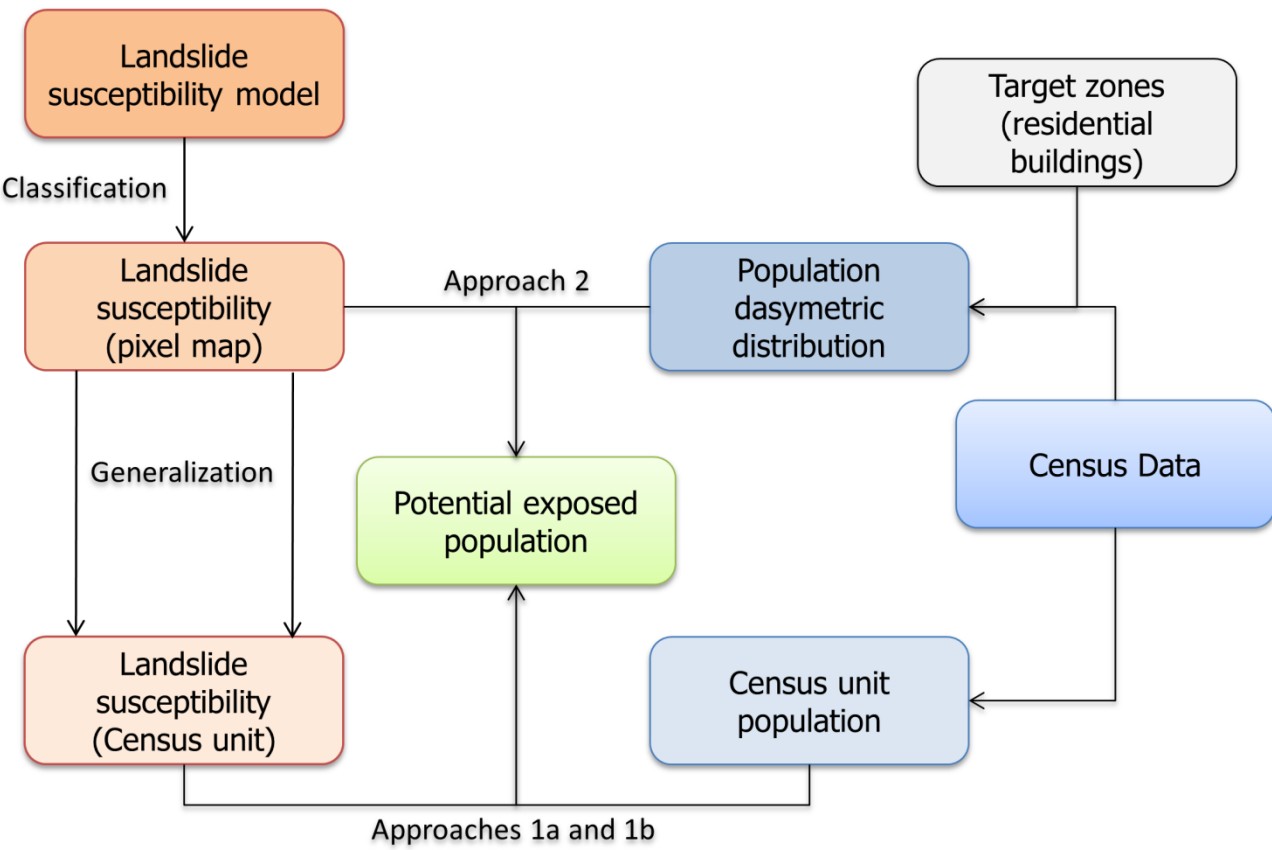

**Figure 2: General methodological work flow.**

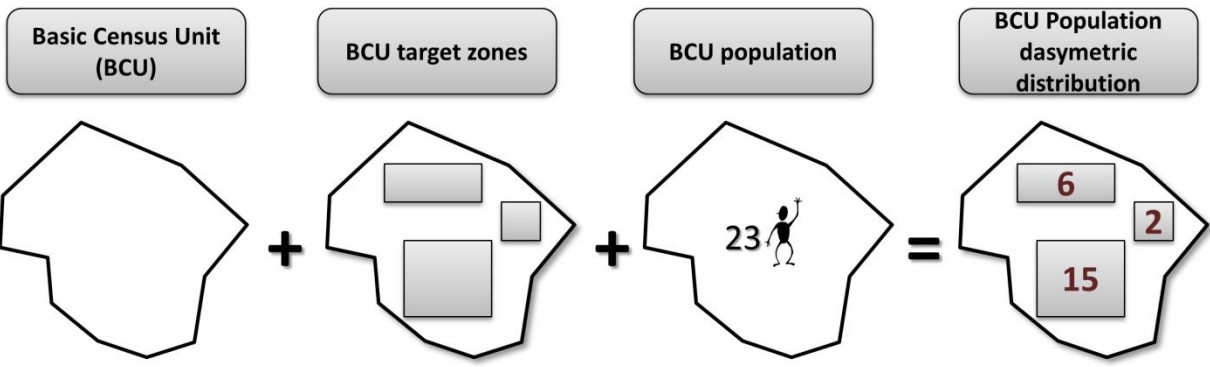

**Figure 3: Schematic dasymetric evaluation of population based on target zone area**

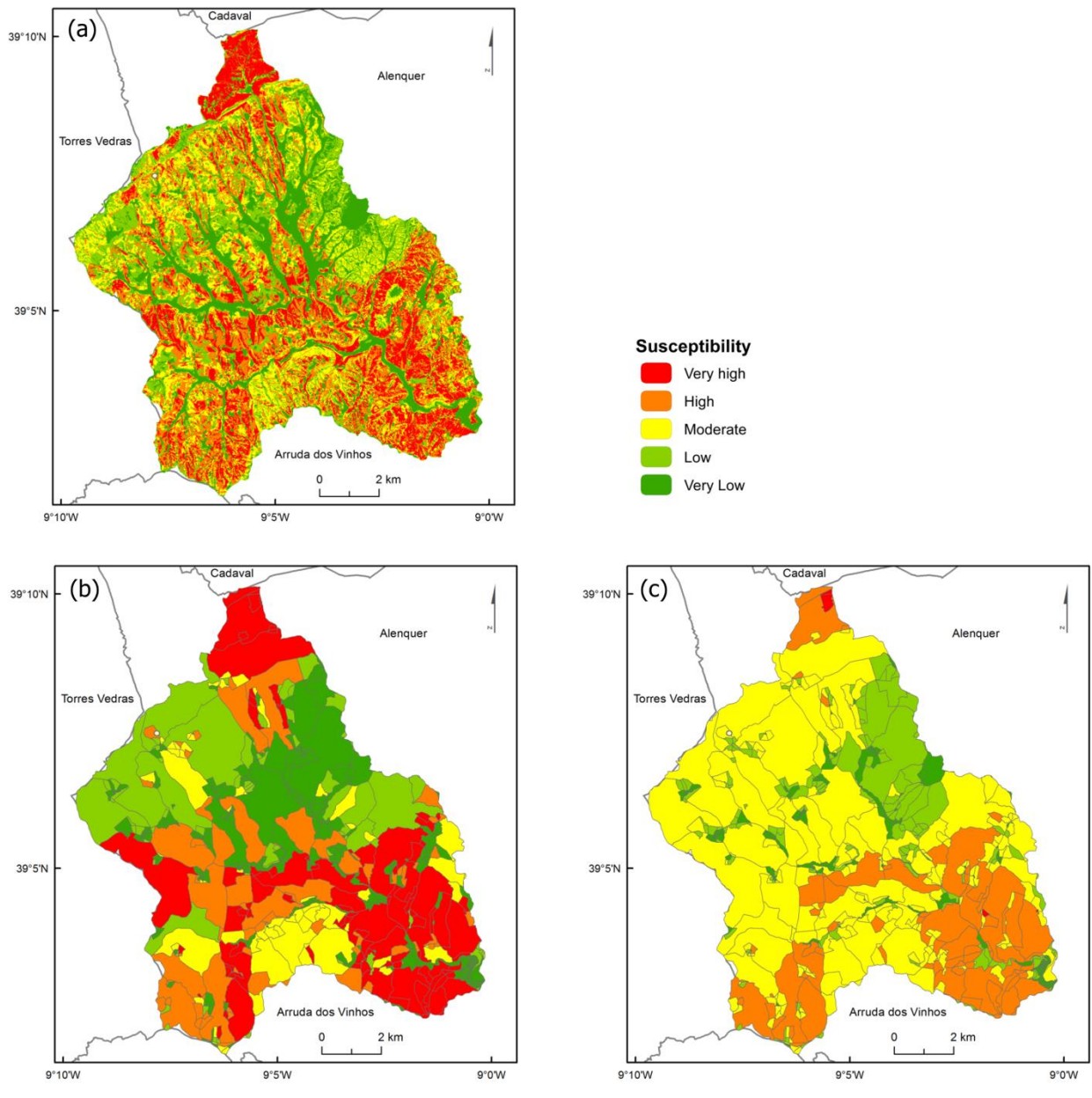

**Figure 4: Deep rotational slides (depth of rupture zone > 3 m) susceptibility maps in Alenquer study area: (a) Approach 2 (pixel-based unit), (b) Approach 1a (Basic Census Unit classified according to the majority susceptibility), (c) Approach 1b (Basic Census Unit classified according to the weighted average of the susceptibility).**

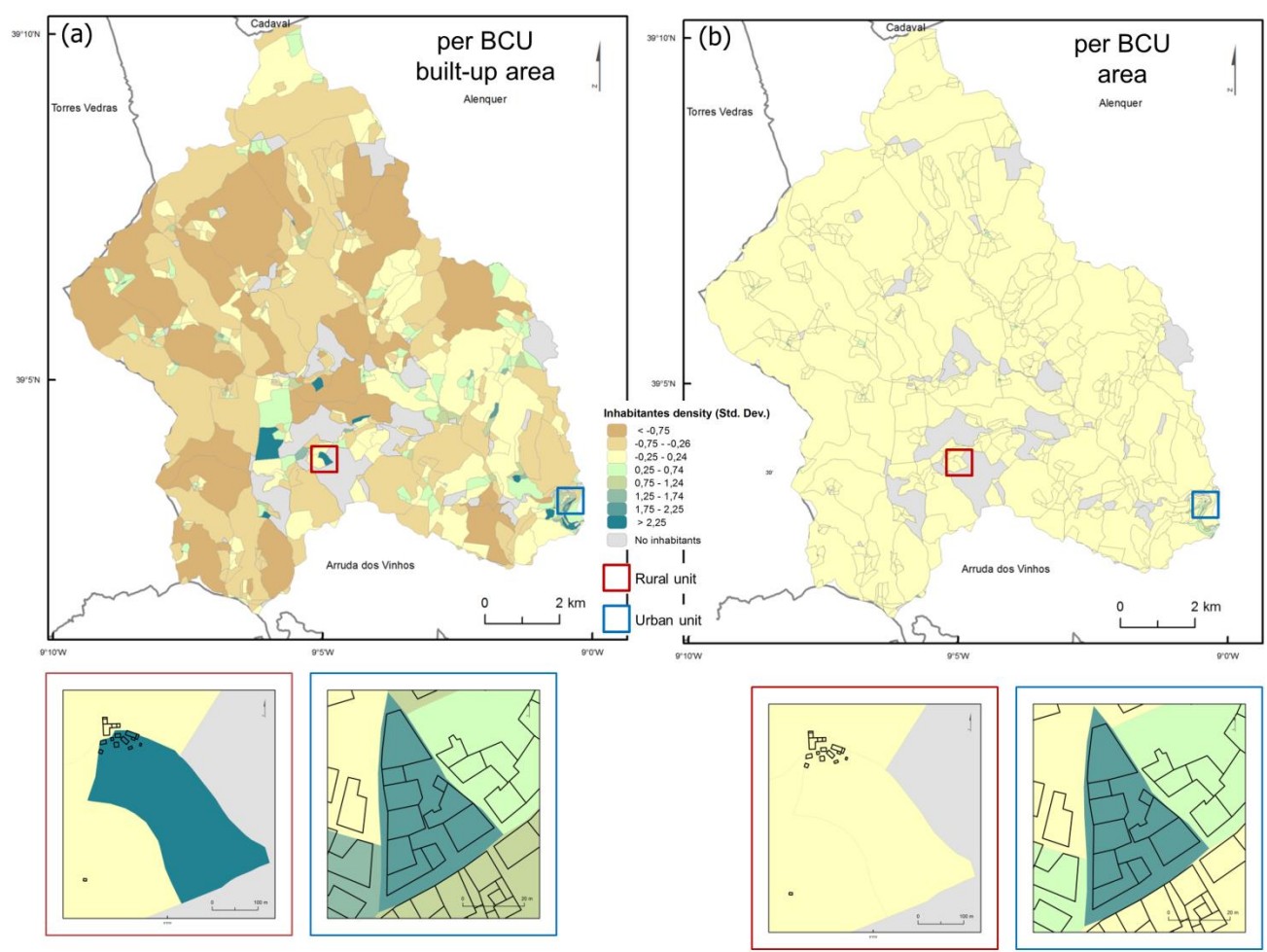

**Figure 5: Population density in the Alenquer study area: (a) per BCU built-up area, (b) per BCU overall area. To facilitate visualization, the classification of target zones in map (a) was extended to the complete BCU.**

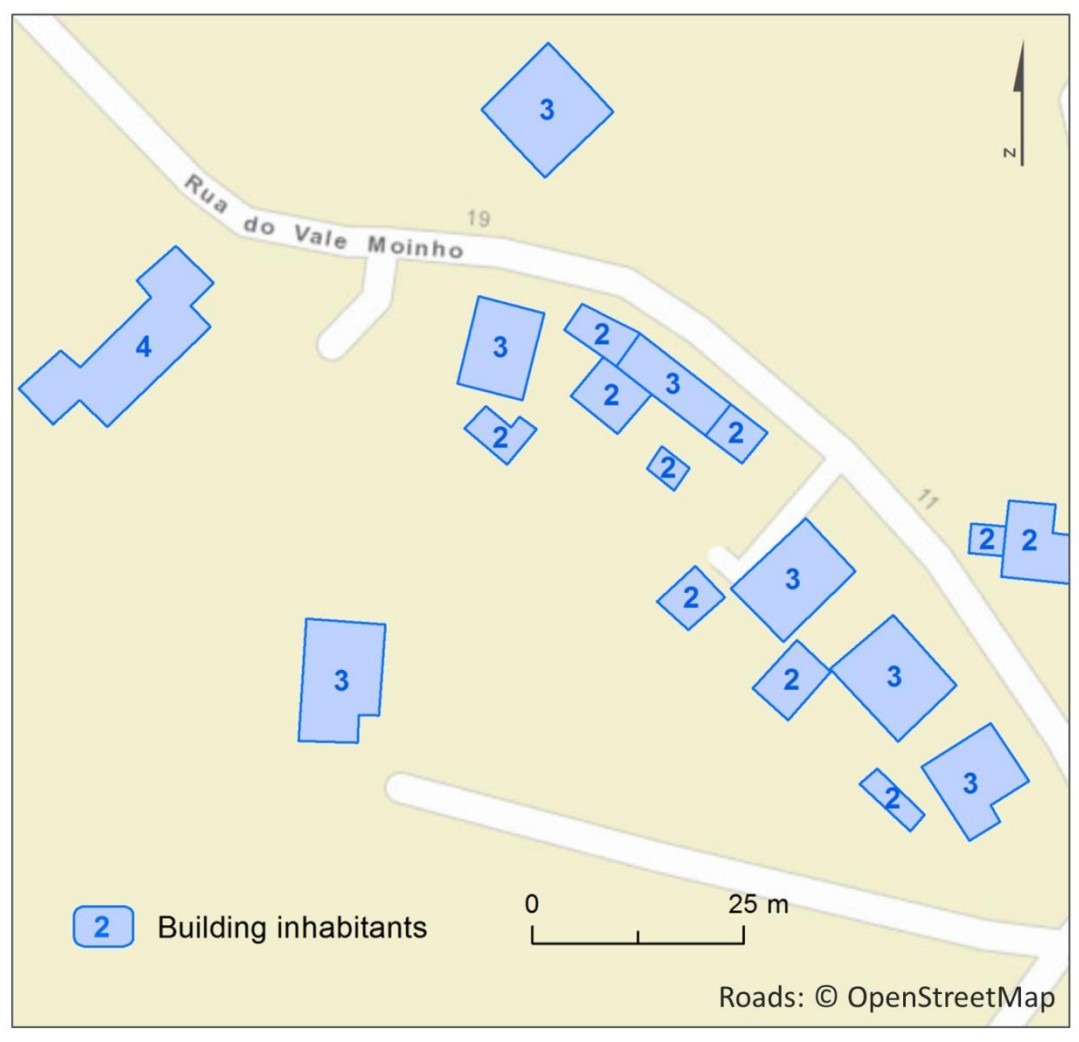

**Figure 6: Potentially exposed population in Alenquer assigned to each building (Approach 2).**

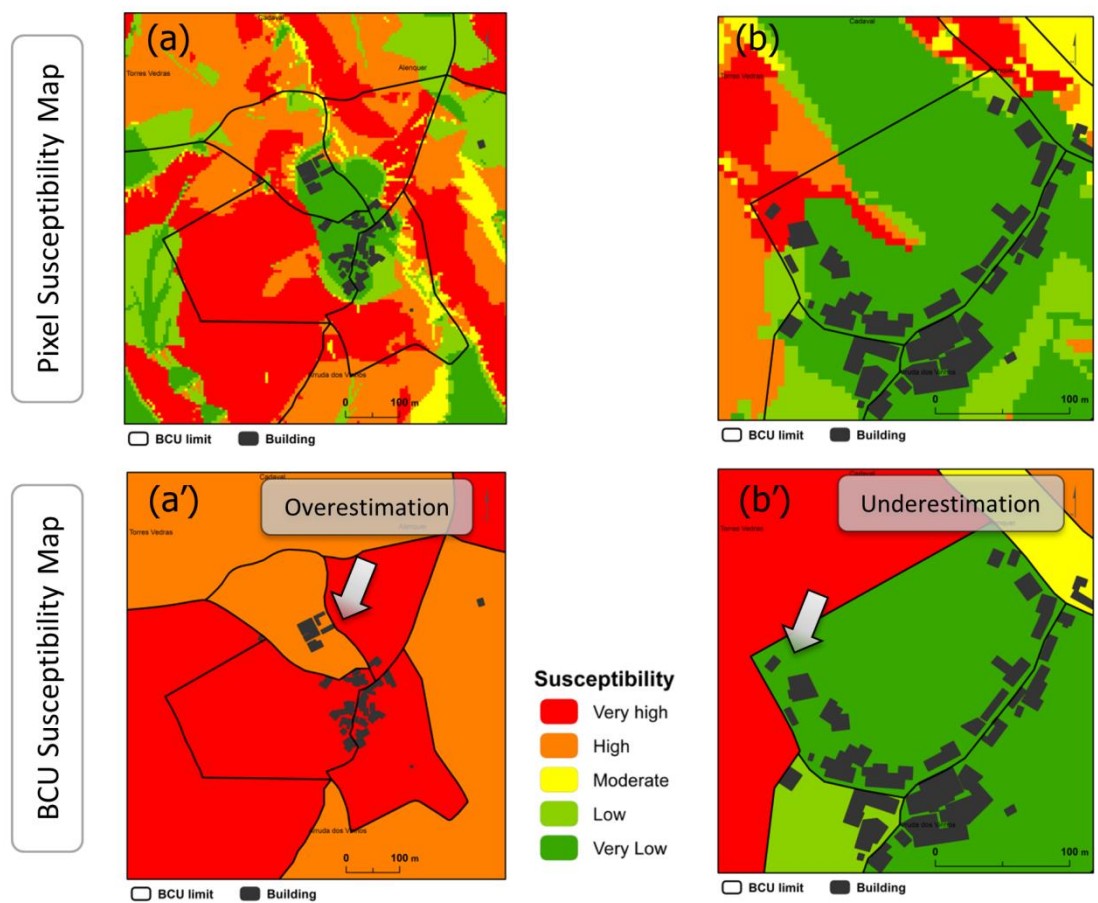

**Figure 7: Examples of overestimation and underestimation of exposed buildings in the Alenquer study area considering the BCU susceptibility map in comparison with the pixel-based susceptibility map.**

