# Peer review of "Assessing population exposure for landslide risk analysis using dasymetric cartography"

_Natural Hazards and Earth System Sciences, 2016_

## Referee Comment (RC1) · M. Papathoma-Koehle (Referee) · 4 Jul 2016

Interactive comment on: "Assessing population exposure for landslide risk analysis using dasymetric cartography" by Garcia R.A.C. et al.

General comments

The authors present a methodology used for dasymetric exposure mapping of population applied in Portugal that can be used from emergency managers to guide evacuation and rescue operations. The disaggregation of population data in order to get a more realistic picture of the population density (especially during different times of the day and the year) and eventually the exposure is very important for the design of emergency operations. However, the specific does not present the methodology used in a comprehensive way due to poor structure and poor English. The article needs restructuring, rewriting of the discussion session giving emphasis in the assumptions and uncertainties and a final editing from a native speaker who can significantly improve the language. For this reason I do not think that it should be accepted for publication in its present form.

Specific comments

-Abstract: Abbreviations such as BCU (line 12) have to be explained at the beginning -Abstract: it needs rewriting to improve the language. Grammatical mistakes and total lack of punctuation (commas) make the article difficult to read and understand. This is relevant also for the rest of the text. -Introduction: The introduction is disproportionally long in comparison with the other chapters. The authors provide a literature review (which is good) but although they explain thoroughly what risk is they do not do the same for other terms that are often used in the manuscript such as "exposure" or "dasymetric mapping". A good idea would be to divided in sub-chapters (objectives, state of the art etc.) -Study area: Here a new piece of information appers regarding the landslide susceptibility map. Is this done by the authors? (apparently, yes) -Study area: Why are you working in this area? Past events? Consequences? -Methodology: The methodology is not thoroughly explained (not at least in this chapter). The two approaches that you refer to in the following chapters should be explained here (ii). More information on obtained data could also be included here. -Landslide susceptibility: (line 19). Why did you choose this classification method? What implications does this decisions have for the reliability of the results. This and other points should be discussed in the discussion chapter. -Population exposure: (line 27-line 31) The authors explain here what a dasymetric method is. I think this belongs to the methodology chapter. - In the previous two chapters (landslide susceptibility and population exposure) a number of points show up that increase uncertainty and need to be discussed in the discussion chapter. For example: 1. classification od landslide susceptibility 2. Section 3.1, line 22: "The landslide susceptibility classification attributed to each BCU was defined according to the majority landslide susceptibility class represented in the

BCU"-What implications does such an assumption have to the uncertainties related to this study? 3. Criteria for the binary analysis. (residential/non-residential buildings) 4. Weighting: this also belongs in my opinion to the methodology. Who decides on the weighting and using which criteria? This is not clear... 5. Page 6, line 26. "..target zones from vector to raster...". How can this information be used by emergency planners? Wouldn't it be more practical for them to have exposure information per building? -Page 7, lines 23-24, Revise the sentence. It makes no sense. -Discussion: The discussion needs rewriting and strengthening. The authors do refer to limitations and advantages but just superficially. The specific study includes a large number of assumptions and uncertainties and each one of them has to be outlined. The advantages have to be illustrated by "examples" on how the results may be used by the emergency planners. Moreover, many issues are completely ignored (e.g. presence of vulnerable groups: the division between residential/non residential is not thoroughly explained. -The authors need a conclusion chapter, outlining their achievements and describing the future perspectives in the specific field.

Technical corrections

Native speaker editing is in my opinion necessary. There are plenty of grammatical mistakes, inconsistent language (approach 1, approach 2?), mistakes in wording e.g. "study case" (instead of case study), "building limits" instead of building footprint, "people inhabitants etc. and parts that are difficult to understand (e.g. "turn off Lisbon metropolitan area"). The lack of commas makes also the understanding of the text very difficult.

Please also note the supplement to this comment:
http://www.nat-hazards-earth-syst-sci-discuss.net/nhess-2016-202/nhess-2016-202-RC1-supplement.pdf
* * *
[Figure]

2016.

---

## Referee Comment (RC2) · A. Tavares (Referee) · 19 Jul 2016

It is considered that the article is potentially relevant to NHESS journal readers and can constitute a methodological standpoint article. But the way it is presented and discussed makes it a technical note, which reduces the potential relevance can achieve in studies about hazardous processes. The manuscript presents a good introduction, enumerating the importance of analyzing the impacts, with a good state of the art, in which however lacks recent publications made in the Lisbon metropolitan area where the methodology of territorial vulnerability and the risks, have been discussed. On the framework about the methodology for assessing the dasymetric exposure, and the related mapping, this is consistent, although limited in the discussion, which is reflected later in the discussion of the results, made on an incipient form, or based on the uncertainty related with people location inside buildings, which is a curiosity. It is considered

that in relation to the structure the article it is unbalanced, with a long introduction. The presentation of results is scarce and the discussion is done in bullets through synthetic sentences, requiring a deeper discussion. In terms of the graphical elements presented, they have quality and are illustrative, although a summary table that show the comparative results of the two approaches (1 and 2) it was important. About the quality of the edited English, this is limited, with poor formal expressions, so it is suggested a review by a native speaker. We now present some considerations that the authors should note in reviewing the manuscript: 1 - The introduction is written considering multi-hazards concerns, and then the authors have evolved to the landslides exposed population, based on the landslide susceptibility map characteristics. This concerns about a single hazard could be better explained and supported. 2 - It is not clear that the added value resulting from this methodological development using dasymetric cartography, will be applied to the mapping for the emergency management, as suggested in some paragraphs, or will be applied to the risk prevention or spatial planning, as suggested in other sentences. 3 - There is a clear choice for the analysis of the Alenquer river basin. This choice is not discussed, nor its importance in relation to Lisbon. Urban sprawl appears to justify the choice of Alenquer municipality, and then devalued the functions and mobility regarding the centrality of Lisbon. The presentation of the data also highlights the high agricultural and forestry land use and occupation in certain areas, losing the relevance of the research. 4 - Resulting from the application of the methodology it is not clear the relationship between the two approaches and the type of movement, superficial or deep mass movements. It seems that this discussion could increase notably the cartographic results. The severity of the movements and the speed thereof could be also discussed on the basis of the two approaches. 5 - An important aspect to be pointed is that the population assigned to a BCU is only the resident population according to the values of the Census in Portugal. The buildings that are represented seem to include both those who have residential functions as the buildings with services and commercial functions. This disagreement must be discussed and presented their performance for both approaches. We consider

the option using a simplification between residential building/not residential building areas may have conditioned the results. 6 - It makes sense discuss the evaluation of the dasymetric exposure due to the uncertainty, and this in relation to the susceptibility mapping. Still seems relevant explaining the added value with this approach in relation with low and moderate probability process, a logic of large disasters, or with exposure to the high probability events associated with small disasters. 7 - It makes sense to discuss the types of damages associated with buildings. However the cartographic analysis could also considered, nor only the damage in the structure of buildings, but the access to buildings, the infrastructure damages, e.g. on sewerage, water or electricity supply, which requires complementary graphical representation. According to the above it is considered that the authors easily overcome these major suggested revisions, enabling a better understanding of the methodological contribution of the article and its application to other contexts.

---

## Author Comment (AC1) · 14 Sep 2016

Authors Reply to RC1 – M. Papathoma-Koehle Interactive comment on: "Assessing population exposure for landslide risk analysis using dasymetric cartography" by Garcia R.A.C. et al.

1 General comments The authors present a methodology used for dasymetric exposure mapping of population applied in Portugal that can be used from emergency managers to guide evacuation and rescue operations. The disaggregation of population data in order to get a more realistic picture of the population density (especially during different times of the day and the year) and eventually the exposure is very important for the design of emergency operations. However, the specific does not present the methodology used in a comprehensive way due to poor structure and poor English.

The article needs restructuring, rewriting of the discussion session giving emphasis in the assumptions and uncertainties and a final editing from a native speaker who can significantly improve the language. For this reason I do not think that it should be accepted for publication in its present form.

A.Reply: The authors would like to acknowledge the referee for the deep review of the manuscript and by the constructive comments that will contribute to improve our manuscript. All the comments and suggestions will be considered in the new version of the manuscript and will be discussed individually in our reply to referee. In addition, the new version of the manuscript will be subjected to a final editing made by a native speaker.

Specific comments

2 -Abstract: Abbreviations such as BCU (line 12) have to be explained at the beginning

A.Reply: The comment will be taken into consideration and we will remove the acronym from the abstract. The sentence in the abstract will be changed as follows: "…as spatial units, the basic census units that is the more detailed data available for regional studies in Portugal"

3-Abstract: it needs rewriting to improve the language. Grammatical mistakes and total lack of punctuation (commas) make the article difficult to read and understand. This is relevant also for the rest of the text.

A.Reply: The authors understand the reviewer comment and apologize for that. Indeed, we hired a specialized translation service to an English native speaker to review the complete final manuscript in order to avoid spelling and grammatical errors.

4-Introduction: The introduction is disproportionally long in comparison with the other chapters. The authors provide a literature review (which is good) but although they explain thoroughly what risk is they do not do the same for other terms that are often used in the manuscript such as "exposure" or "dasymetric mapping". A good idea

would be to divided in sub-chapters (objectives, state of the art etc.)

A.Reply: We acknowledge the referee comment and suggestion. Therefore, Introduction will be split in sub-chapters to make it clear. Disproportionality with other chapters will be taken into consideration but it will decrease with the increasing size of study area, methodology, results and discussion sections. Additionally, definitions about some frequently used terms will be added/referred in introduction as: Exposed elements- considered as elements present in hazard zones that are thereby subject to potential losses (e.g. UNISDR, 2009) Dasymetric mapping – a mapping technic that use ancillary information to turn into a finer resolution coarser input data Susceptibility – considered as the likelihood of landslide occurrence in a specific area according to terrain conditions (Brabb, 1984)

New references in the manuscript:

Brabb, E. E.: Innovative Approaches to Landslide Hazard and Risk Mapping, In: Proceedings 4th International Symposium on Landslides, Toronto, Canadian Geotechnical Society, 1: 307–323, 1984.

UNISDR: 2009 UNISDR Terminology on Disaster Risk Reduction, Int. Strat. Disaster Reduct., 1–30, doi:978-600-6937-11-3, 2009

5-Study area: Here a new piece of information appears regarding the landslide susceptibility map. Is this done by the authors? (Apparently, yes)

A.Reply: We appreciate the comment. In fact the susceptibility map was developed by the first author in his PhD thesis. A reference to the authorship of the landslide susceptibility model will be made in the new version of the manuscript.

6 -Study area: Why are you working in this area? Past events? Consequences?

A.Reply: The authors acknowledge the reviewer comment. The study area is part of the Alenquer municipality and is located in the area north of Lisbon, known as an important landslide prone area (Zêzere et al., 2008). The option for this study area was

supported by three reasons: 1) landslides incidence; 2) type of urban occupation; and 3) social vulnerability. 1) The study area is located in the north of Lisbon region that is a landslide prone area (Zêzere et al., 2008) and according to the DISASTER database (Zêzere et al., 2014), is one of the most import areas in Portugal, considering landslide damage; 2) Additionally, the study area, presents two types of "urban landuse": small villages with a "dense" urban grid as well as disperse settlements. Once the Census units boundaries where quite influenced by settlements density the presence in the study area of two different kinds of territorial occupations allow the comparison of the proposed methodology in two different urban contexts; 3) Moreover, Mendes et al. (2010) in a social vulnerability study for Portugal at municipal scale evaluate the Alenquer municipality as medium criticality ("...defined as the ensemble of individuals' characteristics and behaviours that may contribute to the system's rupture") and Low capability ("defined as the set of territorial infrastructure that enables the community to react in case of disaster"). With this combination the Alenquer municipally is, theoretically, in the region north of Lisbon affected by landslides, one of the least capable to manage hazard consequences; This information will be inserted in the new version of the manuscript.

New references in the manuscript:

Mendes, J. M., Tavares, A. O., Freiria, S. and Cunha, L.: Social vulnerability to natural and technological hazards: The relevance of scale, in Reliability, Risk and Safety: Theory and Applications, vol. 1, edited by R. Briš, C. Guedes Soares, and S. Martorell, pp. 445–451, Taylor & Francis Group, London. [online] Available from: https://estudogeral.sib.uc.pt/jspui/bitstream/10316/25442/1/JMM Esrel 2010.pdf, 2010.

Zêzere, J. L., Pereira, S., Tavares, A. O., Bateira, C., Trigo, R. M., Quaresma, I., Santos, P. P., Santos, M. and Verde, J.: DISASTER: a GIS database on hydro-geomorphologic disasters in Portugal, Nat. Hazards, 72(2), 503–532, doi:10.1007/s11069-013-1018-y, 2014.

[Figure]

7-Methodology: (i) The methodology is not thoroughly explained (not at least in this chapter). The two approaches that you refer to in the following chapters should be explained here (ii). More information on obtained data could also be included here.

A.Reply: The authors thanks the referee comments but, with due respect, we think that is a little misunderstanding because the two chapters that the referee talk about (landslide susceptibility and population exposure) are in fact sub-chapters (3.1 and 3.2) of the Data and methodology chapter (3). Despite of that we will clarify the complete methodological process. Changes in figure 2 (general methodological approach), scale and source of building maps, and criteria for classification of residential buildings will be added in the new version of the manuscript.

8-Landslide susceptibility: (line 19). Why did you choose this classification method? What implications does this decisions have for the reliability of the results. This and other points should be discussed in the discussion chapter.

A.Reply: The authors totally agree with the referee comment. In fact, the used method to classify susceptibility map as well as the chosen number of classes can change the obtained results, once exposed population will have different distributions per suscep-tibility class. However, the focus of this work is not to compare different classification methods or different number of susceptibility classes on exposition results. Additionally, the option for the classification based on a quantile method aims to get susceptibility classes with similar sizes and thus not under- or over-value the importance of any of those classes. Nevertheless, this source of uncertainty will be referred in Discussion section. In addition, we made a test to evaluate changes in obtained results depending on the generalization from raster to statistical terrain units:1) the majority susceptibil-ity value of each statistical unit; 2) the mean susceptibility value of each statistical unit. This will be included in the new version of the manuscript. Therefore, 3 different susceptibility maps and three different exposed population distributions were obtained. The obtained results reveal that the generalization methods significantly influence the importance of each susceptibility class. However, the advantage of the dasymetric

cartography approach to assess the number of inhabitants still remains a good option. Results and Discussion sections will be added with tables and text about these topics.

9-Population exposure: (line 27-line 31) The authors explain here what a dasymetric method is. I think this belongs to the methodology chapter.

A.Reply: Data and methodology chapter (3) include in fact sub-chapters 3.1 and 3.2 devoted to landslide susceptibility and population exposure methodological information, respectively.

10- In the previous two chapters (landslide susceptibility and population exposure) a number of points show up that increase uncertainty and need to be discussed in the discussion chapter. For example: 1. classification od landslide susceptibility 2. Section 3.1, line 22: "The landslide susceptibility classification attributed to each BCU was defined according to the majority landslide susceptibility class represented in the BCU"-What implications does such an assumption have to the uncertainties related to this study? 3. Criteria for the binary analysis. (residential/non-residential buildings) 4. Weighting: this also belongs in my opinion to the methodology. Who decides on the weighting and using which criteria? This is not clear... 5. Page 6, line 26. "..target zones from vector to raster...". How can this information be used by emergency planners? Wouldn't it be more practical for them to have exposure information per building?

A.Reply: The authors acknowledge referee comments and all these topics will be added to the newer version of the manuscript. 1) this will be discussed in the new version of the manuscript as referred in our previous reply; 2) tests considering two different methods were done, as described in our previous reply; 3) vector building maps have attribute fields that allows differentiating some type of buildings (e.g. police stations, fire stations, schools, court, medical facilities, among others). Additionally, during detailed field work other buildings were identified as storage buildings or factory buildings. However, some buildings could have more than one use. In the present

work all the buildings exclusively residential (93%) or mainly residential (5%) were considered as ancillary information. For the remaining buildings there were not assigned population. This information will be included in the new version of the manuscript. 4) The weighting is inserted on methodology. In the first version of the manuscript was presented a general weighting formula that can be used in dasymetric cartography. However, we recognize that in the present work only building area is available to use as ancillary information to weight the importance of buildings. To make it clear we will remove from methodology the general formula (with several parameters) and any reference to other parameters that can be used to weight target zones. 5) The convertion of the target zones from vector to a grid structure was only made to the assessment of the number of people exposed in each susceptibility class and therefore to easily compare the results obtained with approaches #1 and #2. Of course, we agree this information is not useful to Civil Protection. A map where inhabitants are addressed to each specific building should be provided for Civil Protection end users. This will be discussed and a new figure will be inserted in the new version of the manuscript.

11 -Page 7, lines 23-24, Revise the sentence. It makes no sense.

A.Reply: It will be done, it was a tip.

12-Discussion: The discussion needs rewriting and strengthening. The authors do refer to limitations and advantages but just superficially. The specific study includes a large number of assumptions and uncertainties and each one of them has to be outlined. The advantages have to be illustrated by "examples" on how the results may be used by the emergency planners. Moreover, many issues are completely ignored (e.g. presence of vulnerable groups: the division between residential/non residential is not thoroughly explained.

A.Reply: As written above many of the topics will be added to Discussion section (e.g. influence of the susceptibility classification method, generalizations method, use of not exclusively residential buildings). Although the aim of the present work is only to assess

the number of inhabitants potentially exposed to a specific hazard, the new version of the manuscript will include reference to other topics that significantly influence the real exposure of people to landslide hazard. Topics as degree of people vulnerability due to their characteristics (e.g. mobility, age, education, number of year living on that place, etc.), due to building resistance, access to buildings or access to infrastructures and facilities (e.g. sewerage, water or electricity supply, medical care, etc.) will be included in Discussion section

13- The authors need a conclusion chapter, outlining their achievements and describing the future perspectives in the specific field.

A.Reply: The referee suggestion will be take in account and a Conclusion section will be included in the new version of the manuscript.

14 Technical corrections Native speaker editing is in my opinion necessary. There are plenty of grammatical mistakes, inconsistent language (approach 1, approach 2?), mistakes in wording e.g. "study case" (instead of case study), "building limits" instead of building footprint, "people inhabitants etc. and parts that are difficult to understand (e.g. "turn off Lisbon metropolitan area"). The lack of commas makes also the understanding of the text very difficult.

A.Reply: The authors apologize for those mistakes and a specialized translation to an English native speaker is appointed to review the final version of the manuscript.

---

## Author Comment (AC2) · 14 Sep 2016

Author Reply to RC2 – Alexandre Tavares Interactive comment on: "Assessing population exposure for landslide risk analysis using dasymetric cartography" by Garcia R.A.C. et al.

It is considered that the article is potentially relevant to NHESS journal readers and can constitute a methodological standpoint article. But the way it is presented and discussed makes it a technical note, which reduces the potential relevance can achieve in studies about hazardous processes.

A.Reply: The authors would like to acknowledge the referee for the deep review of the manuscript and by the constructive comments that will contribute to improve the new version of our manuscript. All the comments and suggestions will be considered in the

new version of the manuscript and will be discussed individually in our reply to referee.

a)The manuscript presents a good introduction, enumerating the importance of analyzing the impacts, with a good state of the art, in which however lacks recent publications made in the Lisbon metropolitan area where the methodology of territorial vulnerability and the risks, have been discussed.

A.Reply: The authors completely agree with the referee comment. New references considering vulnerability studies at different scales and different risks will be added in the state of the art section, namely:

Guillard-Gonçalves, C., Cutter, S. L., Emrich, C. T. and Zêzere, J. L.: Application of Social Vulnerability Index (SoVI) and delineation of natural risk zones in Greater Lisbon, Portugal, J. Risk Res., 18(5), 651–674, doi:10.1080/13669877.2014.910689, 2015.

Mendes, J. M., Tavares, A. O., Freiria, S. and Cunha, L.: Social vulnerability to natural and technological hazards: The relevance of scale, in Reliability, Risk and Safety: Theory and Applications, vol. 1, edited by R. Briš, C. Guedes Soares, and S. Martorell, pp. 445–451, Taylor & Francis Group, London. [online] Available from: https://estudogeral.sib.uc.pt/jspui/bitstream/10316/25442/1/JMM Esrel 2010.pdf, 2010.

Tavares, A. O. and Santos, P. P. dos: Re-scaling risk governance using local appraisal and community involvement, J. Risk Res., 17(7), 923–949, doi:10.1080/13669877.2013.822915, 2014.

Tavares, A. O., dos Santos, P. P., Freire, P., Fortunato, A. B., Rilo, A. and Sá, L.: Flooding hazard in the Tagus estuarine area: The challenge of scale in vulnerability assessments, Environ. Sci. Policy, 51, 238–255, doi:10.1016/j.envsci.2015.04.010, 2015.

Tavares, A. O. and Santos, P. P. dos: Re-scaling risk governance using local appraisal and community involvement, J. Risk Res., 17(7), 923–949, doi:10.1080/13669877.2013.822915, 2014.

Tavares, A. O., dos Santos, P. P., Freire, P., Fortunato, A. B., Rilo, A. and Sá, L.: Flooding hazard in the Tagus estuarine area: The challenge of scale in vulnerability assessments, Environ. Sci. Policy, 51, 238–255, doi:10.1016/j.envsci.2015.04.010, 2015.

b) On the framework about the methodology for assessing the dasymetric exposure, and the related mapping, this is consistent, although limited in the discussion, which is reflected later in the discussion of the results, made on an incipient form, or based on the uncertainty related with people location inside buildings, which is a curiosity.

A.Reply: The authors thank the referee comment. The authors will clarify Data and methodology section. Changes will be made in figure 2 (general methodological approach), scale and source of building maps, criteria for classification of residential buildings, adopted methods to classifications/generalization of the susceptibility map, etc.. Additionally we will deeply emphasize on assumptions and uncertainties in the Discussion section.

c) It is considered that in relation to the structure the article it is unbalanced, with a long introduction. The presentation of results is scarce and the discussion is done in bullets through synthetic sentences, requiring a deeper discussion.

A.Reply: The authors acknowledge the referee comment. A restructure of the manuscript will be done. Therefore, Introduction will be split in sub-chapters to make it clear. Disproportionality with other chapters will be taken into consideration but it will decrease with the increasing size of study area (with considerations about the adopted criteria to choose this study area), methodology (as referred in our previous reply), results and discussion sections, (e.g. with a test to evaluate changes in obtained results depending on the generalization from raster to statistical terrain units).

d) In terms of the graphical elements presented, they have quality and are illustrative, although a summary table that show the comparative results of the two approaches (1 and 2) it was important.

A.Reply: The authors totally agree with the referee suggestion. Instead of figure 6 two new tables will be inserted in the new version of the manuscript with the results obtained in the comparison of the different approaches: 1) Landslide susceptibility classes weight (%); 2) Potential exposed population per susceptibility class.

e) About the quality of the edited English, this is limited, with poor formal expressions, so it is suggested a review by a native speaker.

A.Reply: The authors understand the reviewer comment and apologize for that. Indeed, we hired a specialized translation service to an English native speaker to review the complete final manuscript in order to avoid spelling and grammatical errors.

We now present some considerations that the authors should note in reviewing the manuscript:

1 - The introduction is written considering multi-hazards concerns, and then the authors have evolved to the landslides exposed population, based on the landslide susceptibility map characteristics. This concerns about a single hazard could be better explained and supported.

A.Reply: The authors acknowledge the referee comment. Despite the references made in introduction to several hazards in the present work only landslide hazard will be considered. In fact the presented methodology can be applied to other hazards but in this specific case the team worked exclusively in landslides, which is not the only hazard that affects the study area but it is one of the most important. The importance of landslides occurrence and consequences in the north of Lisbon region, where study area is located, will be clearer in the new version of the manuscript.

2 - It is not clear that the added value resulting from this methodological development using dasymetric cartography, will be applied to the mapping for the emergency management, as suggested in some paragraphs, or will be applied to the risk prevention or spatial planning, as suggested in other sentences.

A.Reply: The authors thank the referee comment. We agree that the information as presented by authors is not useful to Civil Protection. A map where inhabitants are addressed to each specific building should be provided for Civil Protection end users. This will be discussed and a new figure will be inserted in the new version of the manuscript. Additionally, sentences that suggest that dasymetric cartography results are useful for spatial planning will be removed.

3 - There is a clear choice for the analysis of the Alenquer river basin. This choice is not discussed, nor its importance in relation to Lisbon. Urban sprawl appears to justify the choice of Alenquer municipality, and then devalued the functions and mobility regarding the centrality of Lisbon. The presentation of the data also highlights the high agricultural and forestry land use and occupation in certain areas, losing the relevance of the research.

A.Reply: The authors acknowledge the reviewer comment. The study area is part of the Alenquer municipality and is located in the area north of Lisbon, known as an important landslide prone area (Zêzere et al., 2008). Despite the importance of urban sprawl and the proximity to Lisbon, that certainly influences territorial land use, the option for this study area was supported by three reasons: 1) landslides incidence; 2) type of urban occupation; and 3) social vulnerability. 1) The study area is located in the north of Lisbon region that is a landslide prone area (Zêzere et al., 2008) and according to the DISASTER database (Zêzere et al., 2014), is one of the most import areas in Portugal, considering landslide damage; 2) Additionally, the study area, presents two types of "urban landuse": small villages with a "dense" urban grid as well as disperse settlements. Once the Census units boundaries where quite influenced by settlements density the presence in the study area of two different kinds of territorial occupations allow the comparison of the proposed methodology in two different urban contexts; 3) Moreover, Mendes et al. (2010) in a social vulnerability study for Portugal at municipal scale evaluate the Alenquer municipality as medium criticality ("...defined as the ensemble of individuals' characteristics and behaviours that may contribute to

the system's rupture") and Low capability ("defined as the set of territorial infrastructure that enables the community to react in case of disaster"). With this combination the Alenquer municipally is, theoretically, in the region north of Lisbon affected by landslides, one of the least capable to manage hazard consequences; This information will be inserted in the new version of the manuscript and text will be rewritten to not overemphasize the importance of agricultural and forestry land use.

4 - Resulting from the application of the methodology it is not clear the relationship between the two approaches and the type of movement, superficial or deep mass movements. It seems that this discussion could increase notably the cartographic results. The severity of the movements and the speed thereof could be also discussed on the basis of the two approaches.

A.Reply: The authors thank the referee comment. The presented work only presents deep rotational slides susceptibility maps. In the study area they are generally slow but with displacements capable to significantly damage structures and consequently requiring people evacuation. To avoid misunderstandings all the references to landslides and susceptibility figure caption will indicate that the landslides are deep rotational slides. Additionally, a reference to the velocity and to the severity of damages caused by landslides will be added to the new version of the manuscript.

5 - An important aspect to be pointed is that the population assigned to a BCU is only the resident population according to the values of the Census in Portugal. The buildings that are represented seem to include both those who have residential functions as the buildings with services and commercial functions. This disagreement must be discussed and presented their performance for both approaches. We consider the option using a simplification between residential building/not residential building areas may have conditioned the results.

A.Reply: The authors acknowledge the referee comment and agree that it is no clear that the type of buildings used as ancillary information are only the ones that have

residential purposes. Vector building maps have attribute fields that allows differentiating some type of buildings (e.g. police stations, fire stations, schools, court, medical facilities, among others). Additionally, during detailed field work other buildings were identified as storage buildings or factory buildings. However, some buildings could have more than one use. In the present work all the buildings exclusively residential (93%) or mainly residential (5%) were considered as ancillary information. For the remaining buildings there were not assigned population and they are not cartographically represented. This information will be included in the new version of the manuscript.

6 - It makes sense discuss the evaluation of the dasymetric exposure due to the uncertainty, and this in relation to the susceptibility mapping. Still seems relevant explaining the added value with this approach in relation with low and moderate probability process, a logic of large disasters, or with exposure to the high probability events associated with small disasters.

A.Reply: The authors thank the referee comment. The main aim of this work is to demonstrate that "dasymetric exposure" can be a good method to increase the reliability of the exposed inhabitants distribution when compared to the statistical units approach. We agree that assessing the number of inhabitants is just a single step in a complete risk analysis, which should contemplate cost-benefits analysis considering, for example, probability-intensity relations. We are confident that the proposed methodology can be useful in both situations: (i) low probability phenomena and high magnitude that can result inhigh level of damages, and (ii) high probability events and lower magnitude that is expected to result in low quantity of affected elements. In both cases, when the output is the number of inhabitants per building, once it can help to increase Civil Protection measures efficiency. In fact, the prioritisation of buildings considering the potential affected inhabitants can help the accuracy of rescue operations. In events that cause generalized damages over a high territorial extension the focus on a specific building could not be so important because a whole region is affected. The exception could be, in low density urbanization areas, the buildings where

a high concentration of people is expected. In low magnitude/high frequency events, local damages gain importance and therefore this approach could be slightly more useful. However, this understanding is completely dependent of the type of process, elements at risk, Civil Protection procedures, among many other factors that can influence emergency management operations. A reference to the practical applicability of the proposed methodology in different probability-intensity scenarios will be done in the new version of the manuscript.

7 - It makes sense to discuss the types of damages associated with buildings. However the cartographic analysis could also considered, nor only the damage in the structure of buildings, but the access to buildings, the infrastructure damages, e.g. on sewerage, water or electricity supply, which requires complementary graphical representation.

A.Reply: Although the aim of the present work is only to assess the number of inhabitants potentially exposed to a specific hazard, the new version of the manuscript will include reference to other topics that significantly influence the real exposure of people to landslide hazard. Topics as degree of people vulnerability due to their characteristics (e.g. mobility, age, education, number of year living on that place, etc.), due to building resistance, access to buildings or access to infrastructures and facilities (e.g. sewerage, water or electricity supply, medical care, etc.) will be included in Discussion section

---

## Author Response (AR1)

**Authors Reply to Editor – Sven Fuchs**

Dear Sven Fuchs,

we are sending the new version of the manuscript considering all the comments made by the two reviewers. This new version of the manuscript was also subjected to a final editing made by an English native speaker in order to avoid spelling and grammatical errors.

We insert additional text in this new version of the manuscript, in the Discussion and Conclusions sections, referring to the added value of the presented methodology to a general and global vulnerability assessment and to the practical use by Civil Protection.

Thank you for your attention,

Yours sincerely

RAC Garcia

**Authors Reply to RC1 – M. Papathoma-Koehle**

2nd version: "Assessing population exposure for landslide risk analysis using dasymetric cartography" by Garcia R.A.C. et al.

All significant changes were marked-up in blue in the new version of the manuscript.

**1 General comments**

The authors present a methodology used for dasymetric exposure mapping of population applied in Portugal that can be used from emergency managers to guide evacuation and rescue operations. The disaggregation of population data in order to get a more realistic picture of the population density (especially during different times of the day and the year) and eventually the exposure is very important for the design of emergency operations. However, the specific does not present the methodology used in a comprehensive way due to poor structure and poor English. The article needs restructuring, rewriting of the discussion session giving emphasis in the assumptions and uncertainties and a final editing from a native speaker who can significantly improve the language. For this reason I do not think that it should be accepted for publication in its present form.

**A.Reply:** The authors acknowledge the referee for the deep review of the manuscript and by the constructive comments that will contribute to improve the manuscript. All the comments and suggestions were considered in the new version of the manuscript and were discussed individually.

In addition, the new version of the manuscript was subjected to a final editing made by a native speaker.

**Specific comments**

2 -Abstract: Abbreviations such as BCU (line 12) have to be explained at the beginning

**A.Reply:** The comment was taken into consideration and we removed the acronym from the abstract. The sentence in the abstract was changed as follows:

"…as spatial units, the basic census units which is the best spatial data disaggregation and detailed information available for regional studies in Portugal."

3-Abstract: it needs rewriting to improve the language. Grammatical mistakes and total lack of punctuation (commas) make the article difficult to read and understand. This is relevant also for the rest of the text.

**A.Reply:** The authors understand the reviewer comment and apologize for that. Indeed, we hired a specialized translation service to an English native speaker to review the complete manuscript in order to avoid spelling and grammatical errors.

4-Introduction: The introduction is disproportionally long in comparison with the other chapters. The authors provide a literature review (which is good) but although they explain thoroughly what risk is they do not do the same for other terms that are often used in the manuscript such as "exposure" or "dasymetric mapping". A good idea would be to divided in sub-chapters (objectives, state of the art etc.)

A.Reply: We acknowledge the referee comment and suggestion. Therefore, Introduction has now 3 sub-chapters:. 1.1 General concepts and framework; 1.2 Assessment of population exposure - state of the art and 1.3 Objectives

Disproportionality with other chapters was taken into consideration but it decreased with the increase size of sections dedicated to the study area, methodology, results and discussion. In addition, we add a conclusion section.

Moreover, some relevant definitions were added in introduction, namely:

Exposed elements- "…elements present in hazardous zones that are thereby subject to potential losses (e.g. UNISDR, 2009)

Dasymetric mapping – "which is a cartographic technique that uses ancillary information to increase the resolution of the coarser input data"

Susceptibility – "…considered as the likelihood of landslide occurrence in a specific area according to terrain conditions (Brabb, 1984)"

New references in the manuscript:

Brabb, E. E.: Innovative Approaches to Landslide Hazard and Risk Mapping, In: Proceedings 4th International Symposium on Landslides, Toronto, Canadian Geotechnical Society, 1: 307–323, 1984.

UNISDR: 2009 UNISDR Terminology on Disaster Risk Reduction, Int. Strat. Disaster Reduct., 1–30, doi:978-600-6937-11-3, 2009

5-Study area: Here a new piece of information appears regarding the landslide susceptibility map. Is this done by the authors? (Apparently, yes)

**A.Reply:** We appreciate the comment. In fact the susceptibility map was made by the first author in his PhD thesis. A reference to the authorship of the landslide inventory, landslide susceptibility mapand modelling process was done in the new version of the manuscript (sections 1.3. and 3.1)

6 -Study area: Why are you working in this area? Past events? Consequences?

**A.Reply:** The authors acknowledge the reviewer comment. Study area section includes the following new text.

"The choice of this study area was based on three reasons: i) landslides incidence; ii) type of urban occupation; and iii) social vulnerability.

i) The study area is located to the north of the Lisbon region that is a landslide prone area (Zêzere et al., 2008) and according to the DISASTER database (Zêzere et al., 2014) is one of the areas in Portugal that has sustained severe landslide damage. The present work focuses only on deep rotational slides (depth of rupture zone > 3 m). These landslides are generally slow but encompass horizontal displacements capable to significantly damage structures (e.g. houses) and consequently entail evacuation of people (Garcia, 2012);

ii) The study area presents two types of "urban landuse": small villages with a dense urban grid and disperse settlements. The Census units boundaries were influenced by settlements density, therefore the existence of two different types of territorial occupation in the study area allows the comparison of the proposed methodology applied to two different "urban" contexts;

iii) The study area is, theoretically, one of the least prepared to deal with landslide consequences within the region north of Lisbon. According to Mendes et al. (2010) that evaluated the social vulnerability at the municipal scale in Portugal, the Alenquer municipality has a medium criticality ("defined as the ensemble of individuals' characteristics and behaviours that may contribute to the system's rupture") and low capability ("defined as the set of territorial infrastructures that enables the community to react in case of disaster")."

New references in the manuscript:

Mendes, J. M., Tavares, A. O., Freiria, S. and Cunha, L.: Social vulnerability to natural and technological hazards: The relevance of scale, in Reliability, Risk and Safety: Theory and Applications, vol. 1, edited by R. Briš, C. Guedes Soares, and S. Martorell, pp. 445–451, Taylor & Francis Group, London. [online] Available from: https://estudogeral.sib.uc.pt/jspui/bitstream/10316/25442/1/JMM Esrel 2010.pdf, 2010.

Zêzere, J. L., Pereira, S., Tavares, A. O., Bateira, C., Trigo, R. M., Quaresma, I., Santos, P. P., Santos, M. and Verde, J.: DISASTER: a GIS database on hydro-geomorphologic disasters in Portugal, Nat. Hazards, 72(2), 503–532, doi:10.1007/s11069-013-1018-y, 2014.

7-Methodology: (i) The methodology is not thoroughly explained (not at least in this chapter). The two approaches that you refer to in the following chapters should be explained here (ii). More information on obtained data could also be included here.

**A.Reply:** The authors thanks the referee comments but we think that is a little misunderstanding because the two chapters that the referee talk about (landslide susceptibility and population exposure) are in fact sub-chapters (3.1 and 3.2) of the Data and methodology chapter (3). This remains unchanged in the new version of the manuscript.

Anyway we clarified the methodological process in the new version of the manuscript. Changes in figure 2 (general methodological approach), scale and source of building maps, and criteria for classification of residential buildings were added in the new version of the manuscript.

New text (c.f. Sect. 3.2): "The building layer (1:10 000 vector map from Alenquer Municipality) has attribute fields that allows differentiating the type of services and

commercial buildings (e.g. police stations, fire stations, schools, court, medical facilities, among others). Additionally, during detailed field work the non-residential buildings were identified, e.g. storage buildings, factory buildings, and that information was added to the original database. All the other buildings were regarded as intended for residential use. However, some buildings could have more than one function. In the present work all the buildings that were exclusively residential or mainly residential were considered as ancillary information. The remaining buildings were not considered as target zones and they were not assigned any population."

8-Landslide susceptibility: (line 19). Why did you choose this classification method? What implications does this decisions have for the reliability of the results. This and other points should be discussed in the discussion chapter.

**A.Reply:** The authors totally agree with the referee comment. In fact, the used method to classify susceptibility map as well as the chosen number of classes can change the obtained results, once exposed population will have different distributions per susceptibility class. However, the focus of this work is not to compare different classification methods or different number of susceptibility classes on exposure results. Additionally, the option for the classification based on a quantile method aims to get susceptibility classes with similar sizes and thus not under- or over-value the importance of any class. Nevertheless, the classification method is explained in the methodology section and this source of uncertainty was referred in the Discussion section in the new version of the manuscript.

Methodology: "The option for the classification based on a quantile method aims to get susceptibility classes with similar size without assigning importance a priori to any of those classes based on their sizes. However, to use census unit maps a susceptibility value should be addressed to each BCU. So, in a subsequent step the classified pixel-based landslide susceptibility map was overlaid to the BCU map (vector structure), and a landslide susceptibility classification attributed to each BCU. The generalization was made using two different techniques: i) the BCU susceptibility class was defined according to the majority landslide susceptibility class presented in the BCU; ii) the overall susceptibility of the BCU is the weighted average of identified susceptibility classes."

Discussion: "Independently of the approach some uncertainties are present and affect the obtained results. In fact, the classification of the "original" landslide susceptibility

map (pixel-based) is needed to generate landside susceptibility maps based on statistical units. The number of classes and the chosen method to generalize the susceptibility may produce differences in the obtained results once the range of classes may be higher and the importance of each class become significantly different, which influences the number of inhabitants in each class. However, the focus of this work is not to evaluate how classification methods or different number of susceptibility classes influence the assessment of exposed inhabitants. Hence the option for the classification based on a quantile method that aims to get susceptibility classes with similar size without assigning importance a priori to any of those classes based on their sizes. Moreover, the number of persons in each susceptibility class is only used to compare the adopted approaches. Even though the number of people per susceptibility class can change, it does not affect the distribution of people per buildings that is independent of the number of susceptibility classes."

In addition, we made a test to evaluate changes in obtained results depending on the generalization from raster to statistical terrain units:1) the majority susceptibility value of each statistical unit; 2) the mean susceptibility value of each statistical unit. This was also included in the new version of the manuscript. Therefore, 3 different susceptibility maps and three different exposed population distributions were obtained. The obtained results reveal that the generalization methods significantly influence the importance of each susceptibility class. However, the dasymetric cartography approach to assess the number of inhabitants still remains a good option. Results and Discussion sections were added with tables and text about these topics.

9-Population exposure: (line 27-line 31) The authors explain here what a dasymetric method is. I think this belongs to the methodology chapter.

**A.Reply:** Data and methodology chapter (3) include in fact sub-chapters 3.1 and 3.2 devoted to landslide susceptibility and population exposure methodological information, respectively.

10- In the previous two chapters (landslide susceptibility and population exposure) a number of points show up that increase uncertainty and need to be discussed in the discussion chapter. For example:

1. classification od landslide susceptibility

2. Section 3.1, line 22: "The landslide susceptibility classification attributed to each BCU was defined according to the majority landslide susceptibility class represented in the BCU"-What implications does such an assumption have to the uncertainties related to this study?

3. Criteria for the binary analysis. (residential/non-residential buildings)

4. Weighting: this also belongs in my opinion to the methodology. Who decides on the weighting and using which criteria? This is not clear…

5. Page 6, line 26. "..target zones from vector to raster…". How can this information be used by emergency planners? Wouldn't it be more practical for them to have exposure information per building?

**A.Reply:** The authors acknowledge the referee comments and all these topics were added to the newer version of the manuscript.

1) this topic was discussed in the new version of the manuscript as referred in our previous reply;

2) tests considering two different methods were done, as described in our previous reply;

3) vector building maps have attribute fields that allows differentiating some type of buildings (e.g. police stations, fire stations, schools, court, medical facilities, among others). Additionally, during detailed field work other buildings were identified as storage buildings or factory buildings. However, some buildings could have more than one use. In the present work all the buildings exclusively residential (93%) or mainly residential (5%) were considered as ancillary information. For the remaining buildings there were not assigned population. This information was included in the new version of the manuscript.

New text (c.f. Sect. 3.2): "The building layer (1:10 000 vector map from Alenquer Municipality) has attribute fields that allows differentiating the type of services and commercial buildings (e.g. police stations, fire stations, schools, court, medical facilities, among others). Additionally, during detailed field work the non-residential buildings were identified, e.g. storage buildings, factory buildings, and that information was added to the original database. All the other buildings were regarded as intended for residential use. However, some buildings could have more than one function. In the present work all the buildings that were exclusively residential or mainly residential

were considered as ancillary information. The remaining buildings were not considered as target zones and they were not assigned any population."

4) The weighting is inserted on methodology. In the first version of the manuscript was presented a general weighting formula that can be used in dasymetric cartography. However, we recognize that in the present work only building area is available to use as ancillary information to weight the importance of buildings. To make it clear we remove from methodology section the general formula (with several parameters) and any reference to other parameters that can be used to weight target zones.

5) The conversion of the target zones from vector to a grid structure was only made to assess the number of people exposed in each susceptibility class and therefore to easily compare the results obtained with approaches #1 and #2. Of course, we agree this information is not useful to Civil Protection. A map where inhabitants are addressed to each specific building should be provided for Civil Protection end users. This was discussed in text and a new figure 6 was inserted in the new version of the manuscript.

New text (c.f. Sect. 3.2): "The conversion of the target zones from vector to a grid structure was only made to assess the number of people exposed in each susceptibility class and therefore to easily compare the results obtained among approaches. So, for practical purposes the population is assigned to each building and not to a pixel."

11 -Page 7, lines 23-24, Revise the sentence. It makes no sense.

**A.Reply:** It was done, it was a fault in the text.

12-Discussion: The discussion needs rewriting and strengthening. The authors do refer to limitations and advantages but just superficially. The specific study includes a large number of assumptions and uncertainties and each one of them has to be outlined. The advantages have to be illustrated by "examples" on how the results may be used by the emergency planners. Moreover, many issues are completely ignored (e.g. presence of vulnerable groups: the division between residential/non residential is not thoroughly explained.

**A.Reply:** As written above many of the topics were added to Discussion section (e.g. influence of the susceptibility classification method, generalizations method, use of not exclusively residential buildings) which was deeply revised.

Although the aim of the present work is only to assess the number of inhabitants potentially exposed to a specific hazard, the new version of the manuscript include reference to other topics that significantly influence the real exposure of people to landslide hazard. Topics as degree of people vulnerability due to their characteristics (e.g. mobility, age, education, number of year living on that place, etc.), due to building resistance, access to buildings or access to infrastructures and facilities (e.g. sewerage, water or electricity supply, medical care, etc.) were included in Discussion section

13- The authors need a conclusion chapter, outlining their achievements and describing the future perspectives in the specific field.

**A.Reply:** The referee suggestion was taken into account and a Conclusion section was included in the new version of the manuscript.

**14 Technical corrections** Native speaker editing is in my opinion necessary. There are plenty of grammatical mistakes, inconsistent language (approach 1, approach 2?), mistakes in wording e.g. "study case" (instead of case study), "building limits" instead of building footprint, "people inhabitants etc. and parts that are difficult to understand (e.g. "turn off Lisbon metropolitan area"). The lack of commas makes also the understanding of the text very difficult.

**A.Reply:** The authors apologize for those mistakes and a specialized translation and review the new version of the manuscript was done by an English native speaker.

**Author Reply to RC2 – Alexandre Tavares**

2nd version: "Assessing population exposure for landslide risk analysis using dasymetric cartography" by Garcia R.A.C. et al.

All significant changes were marked-up in blue in the new version of the manuscript.

It is considered that the article is potentially relevant to NHESS journal readers and can constitute a methodological standpoint article. But the way it is presented and discussed makes it a technical note, which reduces the potential relevance can achieve in studies about hazardous processes.

**A.Reply:** The authors thank the referee for the deep review of the manuscript and by the constructive comments that contribute to improve the new version of the manuscript. All comments and suggestions were considered in the new version of the manuscript and are discussed individually.

a)The manuscript presents a good introduction, enumerating the importance of analyzing the impacts, with a good state of the art, in which however lacks recent publications made in the Lisbon metropolitan area where the methodology of territorial vulnerability and the risks, have been discussed.

**A.Reply:** The authors completely agree with the referee comment. New references considering vulnerability studies at different scales and different risks in Portugal were added in the state of the art section, namely:

Guillard-Gonçalves, C., Cutter, S. L., Emrich, C. T. and Zêzere, J. L.: Application of Social Vulnerability Index (SoVI) and delineation of natural risk zones in Greater Lisbon, Portugal, J. Risk Res., 18(5), 651–674, doi:10.1080/13669877.2014.910689, 2015.

Mendes, J. M., Tavares, A. O., Freiria, S. and Cunha, L.: Social vulnerability to natural and technological hazards: The relevance of scale, in Reliability, Risk and Safety: Theory and Applications, vol. 1, edited by R. Briš, C. Guedes Soares, and S. Martorell, pp. 445–

451, Taylor & Francis Group, London. [online] Available from: https://estudogeral.sib.uc.pt/jspui/bitstream/10316/25442/1/JMM Esrel 2010.pdf, 2010.

Santos, P. P. dos, Tavares, A. O. and Zêzere, J. L.: Risk analysis for local management from hydro-geomorphologic disaster databases, Environ. Sci. Policy, 40, 85–100, doi:10.1016/j.envsci.2013.12.007, 2014.

Tavares, A. O. and Santos, P. P. dos: Re-scaling risk governance using local appraisal and community involvement, J. Risk Res., 17(7), 923–949, doi:10.1080/13669877.2013.822915, 2014.

Tavares, A. O., dos Santos, P. P., Freire, P., Fortunato, A. B., Rilo, A. and Sá, L.: Flooding hazard in the Tagus estuarine area: The challenge of scale in vulnerability assessments, Environ. Sci. Policy, 51, 238–255, doi:10.1016/j.envsci.2015.04.010, 2015.

b) On the framework about the methodology for assessing the dasymetric exposure, and the related mapping, this is consistent, although limited in the discussion, which is reflected later in the discussion of the results, made on an incipient form, or based on the uncertainty related with people location inside buildings, which is a curiosity.

**A.Reply:** The authors thank the referee comment. The authors tried to clarify Data and methodology section. Changes were made in figure 2 (general methodological approach), scale and source of buildings map, criteria for classification of residential buildings, and adopted methods to classification/generalization of the susceptibility map.

Additionally we emphasized assumptions and uncertainties related to these topics in the Discussion section.

c) It is considered that in relation to the structure the article it is unbalanced, with a long introduction. The presentation of results is scarce and the discussion is done in bullets through synthetic sentences, requiring a deeper discussion.

**A.Reply:** The authors acknowledge the referee comment. A restructure of the manuscript was done. Therefore, Introduction has now 3 sub-chapters (1.1 General

concepts and framework; 1.2 Assessment of population exposure - state of the art and 1.3 Objectives)

Disproportionality with other chapters were taken into consideration but it decreased with the increasing size of section devoted to study area (with considerations about the adopted criteria to choose this study area), methodology (as referred in our previous reply), results and discussion, and a new conclusion section.

d) In terms of the graphical elements presented, they have quality and are illustrative, although a summary table that show the comparative results of the two approaches (1 and 2) it was important.

A.Reply: The authors totally agree with the referee suggestion. Instead of figure 6 two new tables were inserted in the new version of the manuscript to better compare results obtained with different approaches: 1) Table 1. Landslide susceptibility classes (%) in Alenquer study area; 2) Table 2: Potentially exposed population per susceptibility class in Alenquer study area.

e) About the quality of the edited English, this is limited, with poor formal expressions, so it is suggested a review by a native speaker.

**A.Reply:** The authors understand the reviewer comment and apologize for that. Indeed, an English native speaker did a final review to the new version of the manuscript in order to avoid spelling and grammatical errors.

We now present some considerations that the authors should note in reviewing the manuscript:

1 - The introduction is written considering multi-hazards concerns, and then the authors have evolved to the landslides exposed population, based on the landslide susceptibility map characteristics. This concerns about a single hazard could be better explained and supported.

A.Reply: The authors acknowledge the referee comment. Despite the references made in introduction to several hazards in the present work only landslide hazard is considered. In fact the presented methodology can be applied to other hazards but in this specific case we worked exclusively on landslides, which is not the only hazard that affects the study area but it is one of the most important. The importance of landslides occurrence and consequences in the north of Lisbon region, where the study area is located, was highlighted in the new version of the manuscript.

2 - It is not clear that the added value resulting from this methodological development using dasymetric cartography, will be applied to the mapping for the emergency management, as suggested in some paragraphs, or will be applied to the risk prevention or spatial planning, as suggested in other sentences.

A.Reply: The authors thank the referee comment. We agree that the information as presented by authors is not useful to Civil Protection. A map where inhabitants are addressed to each specific building should be provided for Civil Protection end users. This discussion and a new figure 6 were inserted in the new version of the manuscript. Additionally, sentences that suggest that dasymetric cartography results are useful for spatial planning were removed.

3 - There is a clear choice for the analysis of the Alenquer river basin. This choice is not discussed, nor its importance in relation to Lisbon. Urban sprawl appears to justify the choice of Alenquer municipality, and then devalued the functions and mobility regarding the centrality of Lisbon. The presentation of the data also highlights the high agricultural and forestry land use and occupation in certain areas, losing the relevance of the research.

A.Reply: The authors acknowledge the reviewer comment. Study area section includes now the following text.

"The choice of this study area was based on three reasons: i) landslides incidence; ii) type of urban occupation; and iii) social vulnerability.

i) The study area is located to the north of the Lisbon region that is a landslide prone area (Zêzere et al., 2008) and according to the DISASTER database (Zêzere et al., 2014) is one of the areas in Portugal that has sustained severe landslide damage. The present work focuses only on deep rotational slides (depth of rupture zone > 3 m). These

landslides are generally slow but encompass horizontal displacements capable to significantly damage structures (e.g. houses) and consequently entail evacuation of people (Garcia, 2012);

ii) The study area presents two types of "urban landuse": small villages with a dense urban grid and disperse settlements. The Census units boundaries were influenced by settlements density, therefore the existence of two different types of territorial occupation in the study area allows the comparison of the proposed methodology applied to two different "urban" contexts;

iii) The study area is, theoretically, one of the least prepared to deal with landslide consequences within the region north of Lisbon. According to Mendes et al. (2010) that evaluated the social vulnerability at the municipal scale in Portugal, the Alenquer municipality has a medium criticality ("defined as the ensemble of individuals' characteristics and behaviours that may contribute to the system's rupture") and low capability ("defined as the set of territorial infrastructures that enables the community to react in case of disaster")."

Additionally, we did not overemphasize references to the agricultural land use.

New references in the manuscript:

Mendes, J. M., Tavares, A. O., Freiria, S. and Cunha, L.: Social vulnerability to natural and technological hazards: The relevance of scale, in Reliability, Risk and Safety: Theory and Applications, vol. 1, edited by R. Briš, C. Guedes Soares, and S. Martorell, pp. 445–451, Taylor & Francis Group, London. [online] Available from: https://estudogeral.sib.uc.pt/jspui/bitstream/10316/25442/1/JMM Esrel 2010.pdf, 2010.

Zêzere, J. L., Pereira, S., Tavares, A. O., Bateira, C., Trigo, R. M., Quaresma, I., Santos, P. P., Santos, M. and Verde, J.: DISASTER: a GIS database on hydro-geomorphologic disasters in Portugal, Nat. Hazards, 72(2), 503–532, doi:10.1007/s11069-013-1018-y, 2014.

4 - Resulting from the application of the methodology it is not clear the relationship between the two approaches and the type of movement, superficial or deep mass movements. It seems that this discussion could increase notably the cartographic results. The severity of the movements and the speed thereof could be also discussed on the basis of the two approaches.

A.Reply: The authors thank the referee comment. The present work only deals with deep rotational slides susceptibility maps. In the study area they are generally slow but with displacements capable to significantly damage structures and consequently requiring people evacuation. To avoid misunderstandings all the references to landslides and susceptibility figure caption now indicate that landslides are deep rotational slides. Additionally, a reference to the velocity and to the severity of damages caused by these landslides was added to the new version of the manuscript (c.f. sect. 2).

5 - An important aspect to be pointed is that the population assigned to a BCU is only the resident population according to the values of the Census in Portugal. The buildings that are represented seem to include both those who have residential functions as the buildings with services and commercial functions. This disagreement must be discussed and presented their performance for both approaches. We consider the option using a simplification between residential building/not residential building areas may have conditioned the results.

A.Reply: The authors acknowledge the referee comment and agree that some information is not clear in the previous version of the manuscript. In order to make it clear the following paragraphs were included in the new version of the manuscript.

New text (c.f. sect. 3.2): "The building layer (1:10 000 vector map from Alenquer Municipality) has attribute fields that allows differentiating the type of services and commercial buildings (e.g. police stations, fire stations, schools, court, medical facilities, among others). Additionally, during detailed field work the non-residential buildings were identified, e.g. storage buildings, factory buildings, and that information was added to the original database. All the other buildings were regarded as intended for residential use. However, some buildings could have more than one function. In the present work all the buildings that were exclusively residential or mainly residential were considered as ancillary information. The remaining buildings were not considered as target zones and they were not assigned any population."

New text (c.f. Sect. 5) "The use of Census, as source of population data, requires two major assumptions: i) the resident population does not change in time; ii) people are located at home. These are strong assumptions in the sense that residents are presumed to be at home at all times, and that it does not take into account the fact

that people living outside the study area might actually be in the study area. In fact, this is far from reality because people move around during the day. However, in what concerns the study area there are no data about daily or seasonal fluctuation of population neither at the building scale nor at the considered statistical unit. So, the above scenarios can be considered as the worst case scenarios for the resident people but the fluctuations during day/night to work, school or other outdoor activities should not be neglected."

"The definition of target zones is one source of uncertainty. Therefore a binary classification that takes into account the residential use of the building was done. Despite the fact that the generality of the buildings have their use officially classified in the building layer database and field work validation had been carried out, not all the buildings were individually validated, which is a source of uncertainty. However, we consider that the errors associated to this uncertainty can be neglected due to three reasons: i) the majority of buildings have an exclusively residential use (93%) and the buildings that other than residential use have more than one type of use, are small in number (5% of total buildings); ii) the vast majority of the buildings (96%) in the study area have up to two floors; and iii) once only the area of the building is considered and "double" functions of buildings are confirmed usually in different floors of the building (e.g. ground floor - commercial, 1st floor - residential) the effective area considered as target zone is correct even if the ground floor is not for residential usage. "

6 - It makes sense discuss the evaluation of the dasymetric exposure due to the uncertainty, and this in relation to the susceptibility mapping. Still seems relevant explaining the added value with this approach in relation with low and moderate probability process, a logic of large disasters, or with exposure to the high probability events associated with small disasters.

A.Reply: The authors thank the referee comment.

The main aim of this work was to demonstrate that "dasymetric exposure" can be a good method to increase the reliability of the exposed inhabitants distribution when compared to the statistical units approach. We agree that assessing the number of inhabitants is just a single step in a complete risk analysis, which should contemplate cost-benefits analysis considering, for example, probability-intensity relations. We are confident that the proposed methodology can be useful for Civil Protection in both situations: (i) low probability phenomena and high magnitude that can result in high level of damages, and (ii) high probability events and lower magnitude that is expected

to result in low quantity of affected elements. In fact, the prioritisation of buildings considering the potential affected inhabitants can help the accuracy of rescue operations. In events that cause generalized damages over a high territorial extension the focus on a specific building could not be so important because a whole region is affected. The exception could be, in low density urbanization areas, the buildings where a high concentration of people is expected. In low magnitude/high frequency events, local damages gain importance and therefore this approach could be slightly more useful. However, this understanding is completely dependent of the type of process, elements at risk, Civil Protection procedures, among many other factors that can influence emergency management operations.

A reference to the practical applicability of the proposed methodology in different probability-intensity scenarios was included in the new version of the manuscript.

New text (c.f. sect. 6): "The proposed methodology can be applied in multi-hazard studies and it is useful in both situations considering probability-intensity relations: (i) low probability phenomena and high magnitude that can result in high level of damages, and (ii) high probability events and lower magnitude that are expected to result in few affected elements. In both cases, the estimation of the number of inhabitants per building will be useful to increase the efficiency of actions taken by the Civil Protection. In fact, the prioritisation of buildings bearing in mind the potentially affected inhabitants will enhance the accuracy of rescue operations. In case of events that cause generalized damages over a large territorial extension the focus on a specific building will not be so important because the whole region is affected. The exception can occur in low density urbanization areas and in the buildings where a high concentration of people is expected. In case of magnitude/high frequency events, local damages gain importance and therefore the proposed approach can be more useful. However, this understanding is completely dependent on the type of process, elements at risk, Civil Protection procedures, among many other factors that influence emergency management operations."

7 - It makes sense to discuss the types of damages associated with buildings. However the cartographic analysis could also considered, nor only the damage in the structure of buildings, but the access to buildings, the infrastructure damages, e.g. on sewerage, water or electricity supply, which requires complementary graphical representation.

**A.Reply:** Although the aim of the present work is only to assess the number of inhabitants potentially exposed to a specific hazard, the new version of the manuscript

includes reference to other topics that significantly influence the real exposure of people to landslide hazard. Topics as degree of people vulnerability due to their characteristics (e.g. mobility, age, education, number of year living on that place, etc.), due to building resistance, access to buildings or access to 
[revised manuscript text omitted]

---

## Referee Report (RR1)

**Reviewer's report**

The paper „Assessing population exposure for landslide risk analysis using dasymetric cartography"
by Garcia R.A.C., Oliveira S.C. and Zezere J.L. has been significantly improved and it is recommended
for publication after some minor corrections.

In more detail:

-The abstract reads better highlighting the main points of the paper,

-The language throughout the text is a lot better but, in my opinion it still needs improvement

-The introduction has been improved, shortened and divided into sub-sections which makes it more
comprehensible for the reader,

-The authors have made a significant effort in the presentation of study area which improves the
paper a lot,

-Changes in the figures (figure 2) have been made according to the reviewer's suggestion

-Additions in the discussion chapter and the inclusion of conclusions contributed to an improved
version of the paper.

Please consider the following:

-Correct and clarify some inconsistencies: e.g.

      p.7, lines 8-9: The potentially exposed population to landslide risk was assessed using the
      Census data and two approaches…

      p.10, line6: In this work, three different approaches were used to evaluate the potentially
      exposed inhabitants…

-The language needs still improvement. Typos, grammatical mistakes and lack of punctuation make
the article still difficult to read.

      e.g. p. 13, line 22: "In case of magnitude/high frequency events…"

---

## Author Response (AR2)

Dear Sven Fuchs,

You can find in attachment the new version of the manuscript. All the correction and suggestions of the referees were considered (in blue). Additionally, a final editing of grammar and punctuation was done by an English native speaker professional translator.

5   Yours sincerely

RAC Garcia

[revised manuscript text omitted]